

# Exploring the relationship between avalanche hazard and large-scale terrain choices at a helicopter skiing operation - Insight from run list ratings

Reto Sterchi[1], Pascal Haegeli[1], and Patrick Mair[2]

[1]School of Resource and Environmental Management, Simon Fraser University, Burnaby, BC, V5A 1S6, Canada
[2]Department of Psychology, Harvard University, Cambridge, MA 02138, USA

*Correspondence to*: Pascal Haegeli (pascal_haegeli@sfu.ca)

**Abstract.** While guides in mechanized skiing operations use a well-established terrain selection process to limit their exposure to avalanche hazard and keep the residual risk at an acceptable level, the relationship between the open/closed status of runs
and environmental factors is complex and has so far only received limited attention from research. Using a large data set of over 25 000 operational run list codes from a mechanized skiing operation, we applied a general linear mixed effects model to explore the relationship between acceptable skiing terrain (i.e., status open) and avalanche hazard conditions. Our results show that the magnitude of the effect of avalanche hazard on run list codes depends on the type of terrain that is being assessed by the guiding team. Ski runs in severe alpine terrain with steep lines through large avalanche slopes are much more susceptible
to increases in avalanche hazard than less severe terrain. However, our results also highlight the strong effects of recent skiing on the run coding and thus the importance of prior first-hand experience. Expressing these relationships numerically provides an important step towards the development of meaningful decision aids, which can assist commercial operations to manage their avalanche risk more effectively and efficiently.

## 1 Introduction

The majestic mountains and abundant powder snow make Western Canada a world renown destination for winter backcountry recreation. One of the key players in winter backcountry recreation in Canada is the mechanized skiing industry, where professionally trained guides take paying clients to remote untracked powder slopes using helicopter and snowcats. The industry has been growing since its inception in the 1960s and offers more than 100 000 skier days per winter today (HeliCat Canada, 2016). However, winter backcountry travel is not without risks. Snow avalanches are the most significant hazard
affecting daily operations in mechanized skiing in Canada (Bruns, 1996). Walcher et al. (under review) report that between 1997 and 2016, avalanches accounted for 77% of the overall natural hazard mortality in mechanized skiing in Canada. Operations manage this risk by continuously assessing the local avalanche hazard conditions and carefully choosing appropriate terrain and travel procedures to limit their exposure to avalanche hazard and keep the residual risk at an acceptable level while still providing a high-quality skiing experience.



In Canada, mechanized skiing operations select terrain for skiing by following a well-established, iterative process. This risk management process has been described as a series of filters occurring at multiple spatial and temporal scales (Israelson, 2015) that progressively eliminate skiing terrain from consideration. The daily process starts with a morning meeting where the guiding team assesses the current hazard conditions and produces a large-scale avalanche hazard forecast across the entire

tenure for the day ahead. This hazard assessment is the foundation for the "run list", which represents the first terrain elimination filter. In this step, the guiding team goes through their inventory of predefined ski runs and collectively decides for each run whether it is open or closed for skiing with guests under the expected avalanche hazard conditions. It is important to note that depending on the nature of the operation, the scale of ski runs can range from tightly defined ski lines to areas the size of a medium ski resort. However, regardless of their size, the nature of ski run is consistent enough that they represent

meaningful decision units at this stage of the risk management process. The large-scale, consensus-based run list that emerges from the morning meeting sets the stage for the skiing program of the day. Over the course of a skiing day, terrain choices are further refined and adapted in response to direct field observations. In most helicopter skiing operations, helicopters serve multiple groups of skiers, each of them led by a guide. It is common practice that the guide of the first group serviced by the helicopter (known as the 'lead guide') decides what runs the groups of this helicopter ski. This run choice represents the second

filter in the terrain selection process. The third and final filter of the terrain selection process is the decision of how exactly a particular run is skied, which is the responsibility of the guide of each group. This sequence of (1) run list established by entire guiding team, (2) run choice made by the lead guide and (3) ski line choice within run made by individual guides, highlights the hierarchical and iterative nature of the terrain selection process. At each filter level, the decisions are refined based on increasingly smaller scale avalanche hazard assessments. While avalanche hazard is a critical factor in this process, other

factors such as weather and flying conditions, flight economics, skiing quality, guest preferences and skiing abilities also affect the selection and sequencing of skied terrain (Israelson, 2015). This terrain selection process is repeated every day and guiding teams continuously adjust their terrain choices in response to the observed changes in avalanche hazard conditions.

While the steps of the terrain selection process are well defined and easy to describe, the relationship between environmental factors and terrain selection is complex and has so far only received limited attention from research. Grímsdottír (2004) and

Haegeli (2010) identified critical terrain and avalanche hazard factors contributing to the terrain decisions at the run scale but did not examine the relationship between avalanche hazard conditions and run list codings in more detail. While Hendrikx et al. (2016) and Thumlert and Haegeli (2018) studied the association between small-scale terrain choices and avalanche conditions quantitatively by analyzing patterns in GPS tracks, they did not account for the fact that these choices are embedded in a higher-level, hierarchical and continuous terrain selection process. Having an in-depth, quantitative understanding of each

stage of the terrain selection process is critical for properly tapping into the risk management practices of guiding teams and describing it in a way that offers useful insight into the influencing factors. Only a comprehensive perspective will allow us to capture the existing tacit expertise and extract information on relevant patterns in a way that facilitates learning from the past and developing decision support tools that can aid the terrain selection process in a professional context in a meaningful way.





Furthermore, a quantitative understanding of the professional terrain selection process that properly isolates the effect of avalanche hazard can offer the foundation for the development of terrain guidance for recreationists.

The objective of our study is to advance our understanding of the professional avalanche risk management process by quantitatively examining the relationship between acceptable skiing terrain (i.e., open or closed for guiding) and avalanche

hazard conditions at the run scale using historic avalanche hazard assessments and run list ratings from a commercial helicopter skiing operation.

## 2    Methods

### 2.1    Study site

For this study, we collaborated with Northern Escape Heli Skiing (NEH), a commercial helicopter skiing company based out

of Terrace, BC, Canada (Figure 1). NEH's operating tenure is in the Skeena Mountains and spans an area of nearly 6000 km$^2$. The skiing terrain ranges from 500 m to 2000 m above sea level covering all three elevation bands (alpine, treeline and below treeline). While their entire tenure has 260 established ski runs, much of their skiing is focused on approximately 60 runs in their home drainage, which is the focus of our study. The character of the local snow climate is maritime with storm slab avalanche problems during or immediately following storms being the primary avalanche hazard concerns (McClung and

Schaerer, 2006; Shandro and Haegeli, 2018).

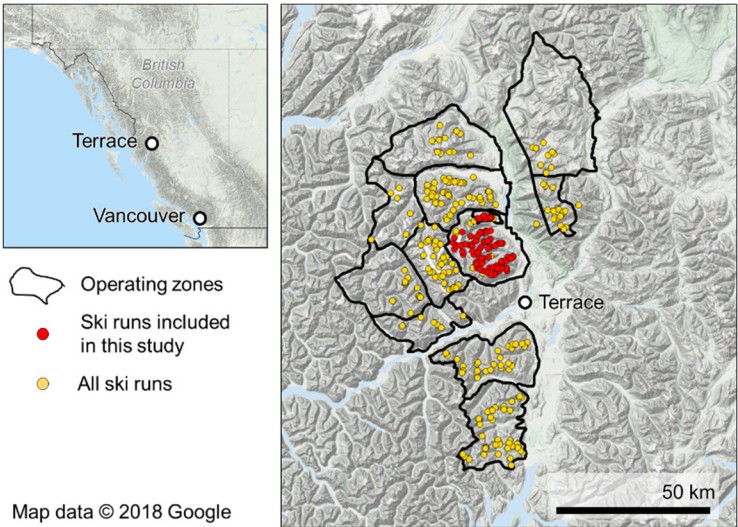

**Figure 1: Geographical overview of the study site with location of the tenure region and the ski runs for one of the operating zones included in**

**this study.**





## 2.2 Data set

The primary dataset used in this study consists of daily run list and avalanche hazard information for the six winter seasons 2012/13 to 2017/18 (517 operational days between December 1 and March 31 of each season). The run list dataset consists of 26 488 individual run ratings in total, one for every run on each of the 517 operational days. At NEH, the guiding team codes runs as either "Open for guiding" (i.e., the run is safe to ski with guests), "Closed for guiding due to avalanche hazard" (i.e., any member of the guiding team is not comfortable with taking guests onto that run), "Closed for guiding for reasons other than avalanche hazard" (e.g. other mountain hazards such as crevasses, open creeks, ski quality) or "Not discussed" (i.e., ski runs in zones not considered are automatically closed for skiing that day).

NEH's avalanche hazard assessment process follows the Conceptual Model of Avalanche Hazard (CMAH, Statham et al., 2018), which provides a framework that structures the process around the identification and characterization of avalanche problems. Avalanche problems represent actual operational concerns about potential avalanches that can be described in terms of the type of avalanche problem, the location in the terrain where the problem can be found, the likelihood of associated avalanches, and their destructive size. The concept of avalanche problem type plays a central role in the CMAH as it represents the idea that distinct types of avalanches that emerge from specific snowpack structures and weather events require different risk mitigation approaches (Statham et al., 2018). Overall, Statham et al. (2018) and describe nine distinct types of avalanches problems (*Dry loose avalanche problem*, *Wet loose avalanche problem*, *Storm slab avalanche problem*, *Wind slab avalanche problem*, *Persistent slab avalanche problem*, *Deep persistent slab avalanche problem, Wet slab avalanche problem*, *Glide avalanche problem*, and *Cornice avalanche problem*) that differ in their development, avalanche activity patterns, how they are best recognized and assessed in the field, and what risk management strategies are most effective for managing them. *Wind slab avalanche problems*, for example, represent cohesive slabs of wind-deposited and broken snow-particles that are typically found on lee-ward (downwind) slopes or in cross-winded areas where winds blow across the terrain. *Wind slab avalanche problems* are relatively easy to manage as the associated avalanches are often limited in size, typically restricted to well-defined terrain features, and tend to stabilize within one or two days after a significant wind event. *Deep persistent slab avalanche problems*, on the other hand, are caused by a thick and hard cohesive slab of snow losing its bond to an underlying weak layer that is deeply buried in the snowpack, often on or near the ground (Haegeli et al., 2010). The formation of *Deep persistent avalanche problems* typically begins in the early season, when conditions are ideal for the development of depth hoar or rain-on-snow events creating facet/crust combinations that are subsequently buried. These weak layers can persist for months and often go dormant with only occasional associated avalanche activity before substantial weather changes in the spring reactivates them again. Since there are often no visible signs of deep persistent slab instability, and associated avalanches tend to be large and essentially not survivable, the management of *Deep persistent slab avalanche problems* is extremely challenging and requires very conservative terrain choices.

After the guides at NEH have identified the types of avalanche problems they are concerned about, they describe the terrain they expect to encounter these problems in terms of elevation bands (alpine, treeline and below treeline) and aspect ranges.





The likelihood of avalanches includes both the sensitivity to triggers and the spatial distribution and is expressed on an ordinal scale using the qualitative terms 'unlikely,' 'possible,' 'likely,' 'very likely' and 'almost certain' (Statham et al., 2018). Destructive size is assessed according to the Canadian avalanche size classification (Canadian Avalanche Association, 2014) on a scale ranging from 1.0 (relatively harmless for people) to 5.0 (largest snow avalanche known for a given path, which

could destroy a village or a large forest area of approximately 40 hectares). Guides express their uncertainty in hazard assessments by specifying ranges of likelihood and size for each avalanche problem (minimum, typical, and maximum for both parameters). The hazard assessments for each elevation band are concluded by summarizing the overall hazard level that emerges from the combined avalanche problems with a single hazard rating on an ordinal scale from 1 to 5. While this hazard scale is derived from the North American Public Avalanche Danger Scale (Statham et al., 2010), it is distinctly different as it

does not include the common signal words (i.e., Low, Moderate, Considerable, High, and Extreme) or travel advice.

To identify meaningful patterns between avalanche hazard and terrain choices numerically, it is critical to encode the nature of the available ski runs in a concise, but insightful way. Avalanche terrain research in the context of backcountry recreation has traditionally primarily focused on standard terrain characteristics such as slope incline, slope shape, elevation, aspect, and vegetation density (e.g., Hendrikx et al., 2016; Thumlert and Haegeli, 2018). More recently, Harvey et al. (2018) developed a

more sophisticated approach that combines an automated identification of potential avalanche release areas with avalanche simulations using RAMMS::EXTENDED (Bartelt et al., 2012; Bartelt et al., 2016) and fall simulations to develop thematic avalanche terrain maps that identify potential avalanche release areas, remote triggering of avalanches, possible runout zones, and the potential of being seriously injured or deeply buried by small or medium-sized avalanches. While the approach by Harvey et al. (2018) offers a much more comprehensive perspective on the nature of avalanche terrain than individual terrain

parameters, the assessment is still at the level of individual raster cells, and it is not completely clear how to combine and summarize these terrain characteristics at the runs scale in a way that fully represents its hazard potential and the overall character of the skiing terrain. Furthermore, in the context of a commercial skiing operation, frequency of use and operational risk management practices (e.g., managing of avalanche hazard through skier compaction or explosive control) also play an important role on whether a particular run is suitable under different hazard conditions, and the overall attractiveness of a run

is further determined by potential access barriers, the general nature of the terrain, the quality of the skiing experience, the operational role of the run and its guidability (Wakefield et al., 2018). These aspects are not only determined by the character of ski runs, but also by the nature of the landing and pickup locations of the run, the operational practices at the operation, and the particular skiing product the operation offers to their clients. To overcome this complexity and include the nature of the ski runs into our model in a way that reflects how professional guides perceive them, we employed the ski run classification

developed by Sterchi and Haegeli (2019). In comparison to existing terrain classification systems with small numbers of universal terrain classes (e.g., ATES; Statham et al., 2006; Campbell and Gould, 2013), Sterchi and Haegeli's approach identifies high-resolution, operation-specific ski run hierarchies based on multi-seasonal patterns in run list ratings (i.e., revealed terrain preferences). Sterchi and Haegeli first identified groups of ski runs by clustering similarly coded ski runs over the course of several winter seasons. Subsequently, they arranged the identified groups into a hierarchy that ranges from runs



that are almost always open to runs that are only open when conditions are favourable. To better understand the nature of the revealed ski run classes, the authors had a senior lead guide at each participating operation provide a comprehensive but structured description of their ski runs with respect to access, type of terrain, skiing experience, operational role, hazard potential, and guide-ability. Since this ski run classification is based on past operational risk management decisions, it reflects

the local terrain expertise and avalanche risk management practices in the context of the available terrain and local snow and avalanche climate conditions (Sterchi and Haegeli, 2019).

At NEH, the analysis of Sterchi and Haegeli (2019) identified six distinct classes of ski runs. To illustrate the nature of the skiing terrain included in this study, Figure 2 shows the average seasonal percentage of run code 'open' for each ski run grouped into the six classes. While the severity of terrain generally increases from Class 1 to Class 6, the groupings also reflect

other run characteristics like accessibility, quality of skiing experience and operational practices.

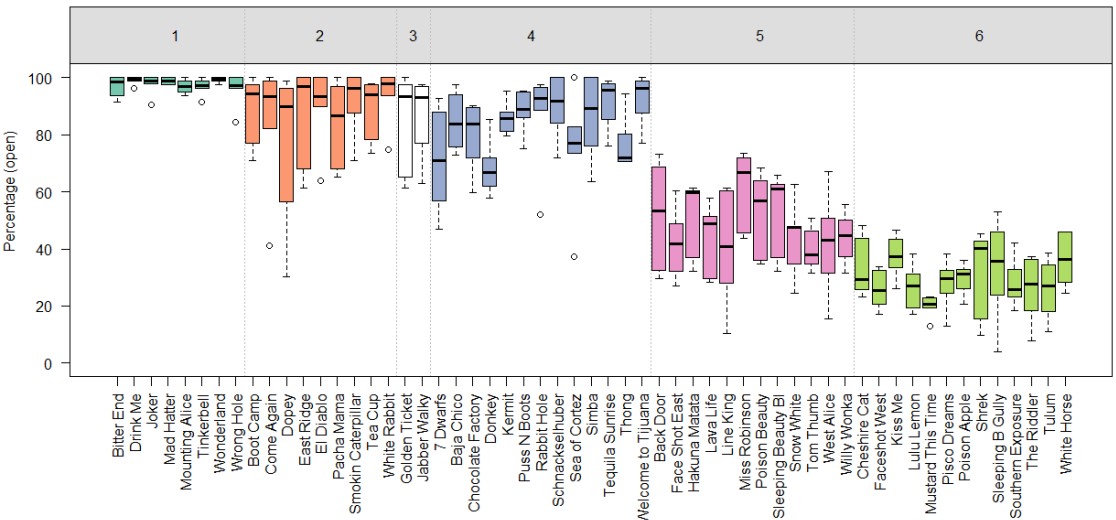

**Figure 2: Average seasonal percentage of run code 'open' for the 57 ski runs during the six seasons 2012/13 to 2017/18 with the six identified classes of similarly managed ski runs (Sterchi & Haegeli, under review). Due to**
**the small group size and their outlier characteristics, the two runs of Class 3 were not included in the present analysis.**

The first three classes generally consist of easily accessible and mostly gentle ski runs with no or only limited exposure to avalanche slopes. Most of the skiing is through open slopes at tree line, open canopy snow forest below tree line, or non-

glaciated or glaciated alpine. The main difference between the first two classes is that the runs of Class 1 provide a better skiing experience. Since Class 1 runs are more attractive, they are typically skied more often, guides have a better handle on the local conditions, and hence the runs are coded open more consistently. The two runs included in Class 3 are of similar general character, but they are located at lower elevations, which makes them more vulnerable to rising freezing levels. Due





to the small group size and their outlier characteristics, we excluded them from the present analysis. While most of the ski runs of the first three groups are at tree line and below, Class 4 to 6 predominantly consist of alpine terrain. Class 4 consists of ski runs in gentle alpine terrain or open slopes at tree line where most ski lines do not cross any avalanche slopes. These ski runs are often accessible and provide generally a good skiing experience with easy or moderately challenging skiing. However, some of the ski runs can be exposed to overhead avalanche hazards during regular avalanche cycles. The ski runs included in Class 5 are also located in the alpine but are substantially steeper and cross avalanche slopes more frequently than the runs of Class 4. Furthermore, almost half of the ski runs in Class 5 can be directly affected by overhead hazard during regular avalanches cycles and many pickup locations are threatened by overhead avalanche hazard during large avalanche cycles. While skiing on these runs was characterized as moderately challenging, they offer very good or even "life-changing" skiing experiences for guests. Class 6, the highest group in the NEH ski run hierarchy, mainly consists of runs in the most serious alpine terrain skied at NEH. The runs are rarely skied but can play an important operational role when conditions are appropriate. Most of these runs have moderately steep or steeper slopes that can produce avalanches of Size 3.0 or bigger and many pickup locations are exposed to overhead avalanche hazard during regular avalanche cycles. However, they provide good or very good skiing experiences for the guests.

## 2.3 Statistical analysis

Since our dataset consists of repeated run list codes for the same runs over the course of several winters, traditional regression models that require observations to be independent from each other are inappropriate for our analysis (Long, 2012). Mixed effects models are an extension of traditional regression models that allow for heterogeneity, nested data, temporal or spatial correlation in longitudinal and/or clustered datasets by relaxing some of the necessary assumptions (Bolker et al., 2009; Zuur et al., 2009; Harrison et al., 2018). To overcome the issue of repeated measures and nested data, mixed effects models include both fixed and random effects in the regression equation. The fixed effects, which are equivalent to the intercept and slope estimates in traditional regression models, capture the relationship between the predictor and response variables for the entire dataset. While traditional regression models assign the remaining unexplained variance in the data (i.e., randomness) entirely to the global error term, mixed effect models partition the unexplained variance that originate from groupings within the dataset into random effects. Thus, random effects can highlight how groups within the dataset deviate from the overall pattern described by the fixed effects. Similar to the parameter estimates for fixed effects, random effects can include both intercept and slope parameters. While random intercepts explain how the average conditions within groups deviate from the average conditions across the entire dataset, random slopes capture group-specific differences in the relationship between the predictor and response variables. The overall response of a particular group to the predictor variables can therefore be described as the linear combination of the overall fixed effects and the group-specific random effects.

Since our target variable, the acceptability of a run, is binary (i.e., open or closed), a logistic regression model is most suited for our analysis. In their basic form, logistic regression models use the logistic function to model the relationship between a



binary dependent variable and one or more predictors $x_i$. In such a model, the probability of $Run_k$ being "open" can be expressed

with $Prob\ (Run_k = "open") = \dfrac{1}{1+e^{-\left(\beta_0 + \Sigma_{i=1}^{j} \beta_i f_i(x_{ik})\right)}}$ .

In this equation, $\beta_0$ is the intercept, $\beta_i$ are the regression parameter estimates associated with the functional forms $f_i()$ (e.g., transformations such as coding a categorical variable into dichotomous variables) of the predictors $x_i$ included in the model.

The linear combination of the predictors $x_{ik}$ multiplied with the parameter estimates $\beta_i$ in the exponent in the denominator represents the log-odds (the logarithm of the odds) of $Run_k$ being "open". The components of the equation can be interpreted as follows: The intercept $\beta_0$ represents the log-odds when all predictors are zero. A parameter estimate of $\beta_i = 1$ or $\beta_i = 2$ means that a one unit increase in $f_i(x_{ik})$ increases the log-odds of $Run_k$ being open by 1 or 2, respectively. This is referred to as the "effect" of the predictor $x_{ik}$. The most common way to express the effect of predictors in logistic regression models is odds

ratios (OR), which can be derived by applying an exponential function to the regression coefficients. Hence, parameter estimates significantly larger than zero result in OR > 1, which means that the odds of $Run_k$ being open increases relative to the base level, whereas parameter estimates significantly smaller than zero produce OR < 1 that highlight that the odds of $Run_k$ being open decreases.

To examine the acceptability of runs (i.e., being open or closed) under different hazard conditions, we regressed their daily

run list codes against the hazard situation with the runs' terrain characteristics, their past use and their run list codes of the previous day as covariates (Figure 3). To focus our analysis on the effect of avalanche hazard on open and closed status of runs, we simplified the categorical run list ratings before fitting the regression model. Run list codes indicating that a run was open (i.e., "Open for guiding") were recoded to 1 whereas run list codes indicating that a run was closed because of avalanche concerns (i.e., "Closed for guiding due to avalanche hazard") were coded as 0. Run list codes indicating that a run was not

considered for any other reasons (i.e., "Closed for guiding for reasons other than avalanche hazard", "Not discussed") were excluded from the analysis.

Avalanche hazard conditions were represented in the model with the *Relevant hazard rating* of the day and the *Types of avalanche problems* present. Since ski runs can cross several elevation bands (e.g., a ski run can start in the alpine, include skiing at treeline and have its pickup location below treeline), multiple avalanche hazard ratings might apply. To circumvent

this issue in our analysis, we derived a *Relevant hazard rating* of the day for each run by taking the highest hazard rating of the elevation bands crossed by the run. *Types of avalanche problem present* was implemented in the model as eight binary covariates (1: present; 0: absent) each representing one of the eight avalanche problems specified by the CMAH. Because the avalanche problems are also assessed for each elevation band separately, we derived relevant daily avalanche problem values for each run similarly to the relevant hazard rating described above. Since avalanches of Size 1.0 to 1.5 are considered relatively

harmless to people (McClung and Schaerer, 2006), we only included avalanche problems in our analysis that were characterized with a maximum destructive size of at least Size 2.0. Because of the small number of cases, we also excluded avalanche problems where the maximum likelihood was assessed lower than "unlikely". To allow our model to account for





the possibility that the effect of avalanche hazard on the acceptability of a run being open might differ among terrain types, we interacted the *Relevant hazard rating* and all eight binary variables for *Types of avalanche problem present* with *Ski Run Class*. To account for the iterative character of the terrain assessment process in mechanized skiing, we included two variables in our model that represent critical temporal influences on run list codes. *Skied in the previous seven days* represents past use, which

offers both first-hand skiing experience and direct weather, snowpack and avalanche observations for a run. *Run code of the previous day* was included to account for the direct influence of previous run lists on subsequent days. To acknowledge possible correlations between *Skied in the previous seven days* and *Run code of the previous day* (i.e., a run needs to be open to be skied) we also added the interaction between these two variables to our model.

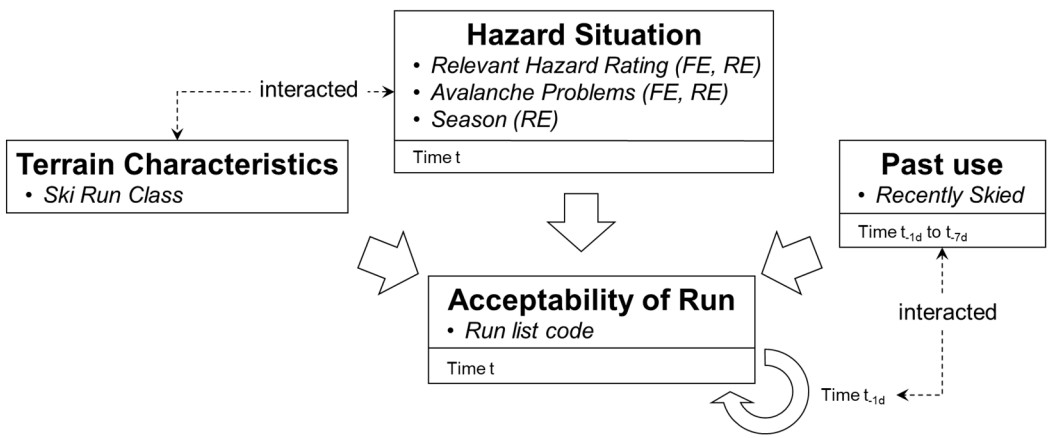

**Figure 3: Illustration of the model. Our model included variables describing the hazard situation, the terrain characteristics of a ski run, and its past use to examine their relationships with the acceptability of a run (e.g., it being coded "open"). To account for the iterative character of the terrain assessment process in mechanized skiing the model also included the run list code from the previous day. In**
**addition to the fixed effects (FE), we included by-run and by-season random effects (RE).**

Since our dataset consists of repeated ratings of the same runs (i.e., panel structure), we included random by-run intercepts and slopes for hazard and avalanche problems. This allows the model to capture the run-specific effect of hazard and avalanche problems goes beyond the ski run class specific effect. We also included a random by-season intercepts to account for the

unique character of each winter in the model.

We performed the model estimation in a Bayesian framework using the statistical software R (R Core Team, 2017) and the package *rstanarm* (Stan Development Team, 2016). We estimated the model with 2500 warmup and 2500 sampling iterations for four separate sampling chains with default priors. Model convergence was inspected based on the potential scale reduction factor (Gelman and Rubin, 1992), which compares the estimated between- and within-chain variances between multiple

Markov chains for each model parameter. Large differences between these variances indicate that a model did not converge while values close to 1.0 indicate good convergence. The Markov chains exhibit some degree of autocorrelation, where a lower autocorrelation indicates more independent sampling of the posterior. The approximate number of independent draws with the





same accuracy as the sample of correlated draws is referred to as the effective sample size (ESS). We consider an ESS of greater than 1000 as an indication of independent sampling of the posterior.

To eliminate the potentially undesirable impact a variable might have purely due to its scale, all variables included in the analysis were scaled to the interval 0 to 1. Hence, *Relevant hazard rating* was included in the model as a numeric variable

scaled to range between 0 and 1. *Ski Run Class* was included as a dummy-coded categorical variable with Class 1 as the reference class, whereas all other predictors were represented as binary variables. We explored different model combinations including models where the avalanche problems of concern were included as categorical variables including combinations of different avalanche problems. Only parameter estimates with 95% credible intervals different from 0 were considered significant.

Since we included both ski run class-specific intercepts and ski run class-specific slopes for hazard ratings, interpreting the effect of avalanche hazard on run list ratings directly from the parameter estimates is challenging. To present the combined effect of intercept and slope, we calculated OR for each ski run class and hazard rating based on the regression coefficients. We present this effect in two tables showing (a) the odds ratios of ski run classes being open with increasing avalanche hazard relative to themselves at hazard *Level 1* and (b) the odds ratios of ski run classes being open with increasing avalanche hazard

relative to ski run Class 1. While the information presented in these two tables are related, they offer slightly different perspectives.

To further illustrate our results and make their interpretation more tangible, we calculated the probabilities of runs of different ski run classes being open under different hazard conditions and operational situations. We present the following three operational scenarios: (a) ski runs were neither open previously nor skied recently, (b) ski runs were not open the day before

but recently skied, and (c) runs were open the day before and recently skied. For each of these scenarios, we plotted the probabilities of ski runs in each ski run class to be open as a function of the hazard rating and included the 50%, 80% and 95% probability intervals based on the averages of 50 draws from the posterior distribution of the individuals runs from each ski run class. Along with the probability curves, average daily percentages of open runs per ski run class are plotted where observations for this scenario existed in the dataset.






## 3    Results and Discussion

The sampling chains of our model converged successfully as indicated by both the potential scale reduction factor (values of
1.0) and for effective sample size (values > 1000) for all parameter estimates. Since the variable *Ski Run Class* was dummy
coded in our model, the main effects for the variables that were interacted with *Ski Run Class* represent the effect for the *Ski
Run Class 1*. The effects for the other classes need to be derived by adding the main effect with the ski run class-specific
interaction effect.

### 3.1    Effect of hazard rating and terrain type

The strongly positive main effect intercept indicates that there is a strong base tendency for the runs of Class 1 to be open at
hazard Level 1 (Table 1). The intercept-ski run class interaction effects for all the other classes are significantly negative (Table
2), which means that overall, they are less likely to be open. As expected, the probability of a run being open decreases
substantially with increasing hazard for all types of terrain as illustrated by the negative main effect for hazard rating (Table
1).

**Table 1: Main effects. Diagnostics and posterior summary statistics of the estimated parameters from the mixed-effects logistic regression model. ESS is the effective sample size for each parameter. Significant parameter estimates are indicated in bold. Not significant (ns) OR omitted.**

| Parameter | Value | ESS | Mean | SD | 2.5% | 97.5% | OR |
|---|---|---|---|---|---|---|---|
| Intercept | - | 2185 | **5.50** | 0.80 | 3.97 | 7.09 | **247.15** |
| Relevant hazard rating | Extreme | 2198 | **-6.59** | 1.12 | -8.79 | -4.40 | **0.001** |
| Deep persistent slab | Present | 2516 | 0.72 | 0.69 | -0.54 | 2.12 | ns |
| Persistent slab | Present | 2956 | 0.1 | 0.45 | -0.77 | 0.98 | ns |
| Storm slab | Present | 2353 | 0.24 | 0.45 | -0.66 | 1.13 | ns |
| Wind slab | Present | 2558 | -0.13 | 0.49 | -1.05 | 0.84 | ns |
| Cornice | Present | 4240 | 1.31 | 1.06 | -0.68 | 3.47 | ns |
| Loose wet avalanche | Present | 3212 | 0.66 | 0.86 | -0.94 | 2.45 | ns |
| Loose dry avalanche | Present | 10000 | -1.14 | 1.95 | -4.90 | 2.66 | ns |
| Wet slab | Present | 4365 | **-1.60** | 0.64 | -2.82 | -0.32 | **0.21** |
| Run code previous day: | Open | 10000 | **2.99** | 0.06 | 2.87 | 3.11 | **19.89** |
| Skied in previous week | Skied | 10000 | **3.44** | 0.42 | 2.64 | 4.29 | **31.19** |

However, the fact that the interaction effects of the different ski run classes (Table 2) differ significantly from each other
highlights that the magnitude of this effect strongly depends on the type of terrain being assessed by the guiding team. These
patterns are also visible in Figure 4, which shows the probabilities of runs of different ski run classes being open for different
hazard ratings and different operational scenarios, but all with a *Storm slab avalanche problem* being a concern. The
visualizations include probability intervals of 50 %, 80 % and 95 % for each ski run class as a whole based on 50 draws from
the posterior distribution. Average daily percentages of open runs per ski run class are plotted as points where observations for
the scenarios exist in the dataset. We can see that the probability of a run being open decreases more substantially with




increasing hazard for runs in Class 5 and 6, whereas the modelled probability curves are less steep for Class 1, 2 and 3 (Figure 4a).

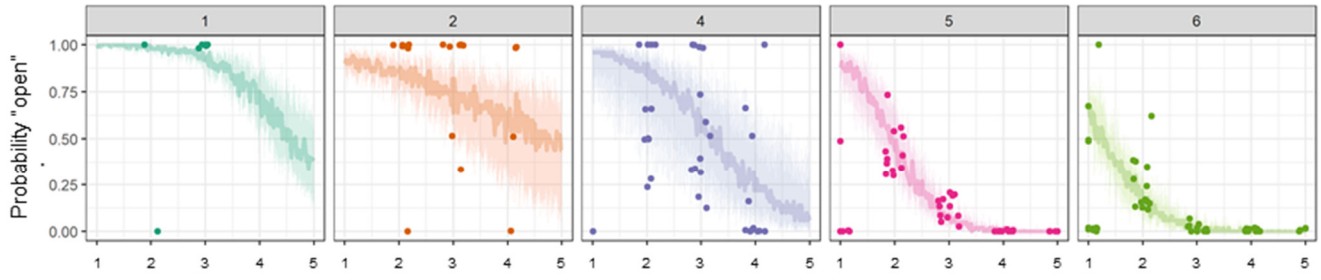

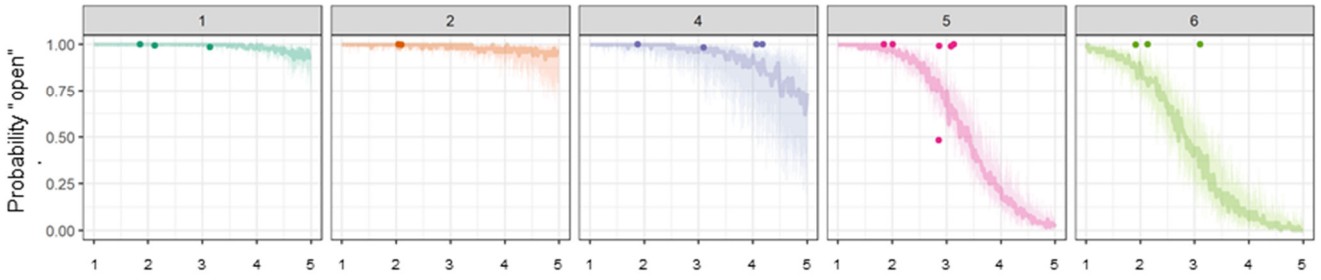

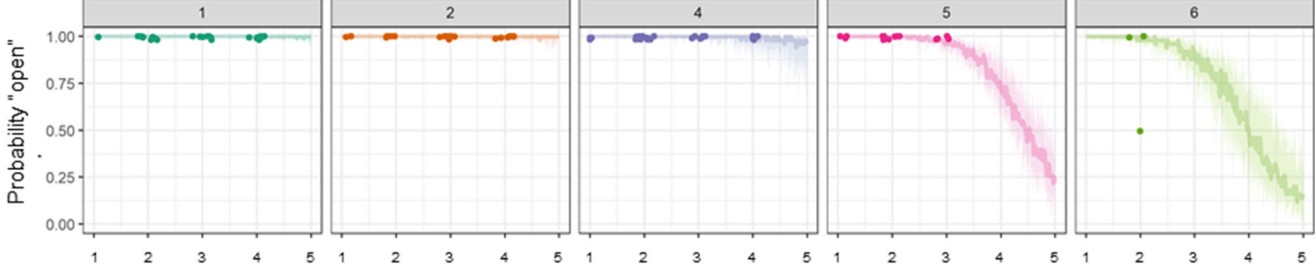

**5** **Figure 4: Probabilities of ski runs being open for Storm slab avalanche problems with (a) a scenario where ski runs were neither open previously nor skied recently, (b) a scenario where runs were not open the day before but recently skied, and (c) a scenario where runs were open the day before and recently skied. The visualizations include probability intervals of 50%, 80% and 95% for each ski run class as a whole based on 50 draws from the posterior distribution. Average daily percentages of open runs per ski run class are plotted as points where observations for this scenario exist in the dataset.**





**Table 2: Interaction effects. Diagnostics and posterior summary statistics of the estimated parameters from the mixed-effects logistic regression model. ESS is the effective sample size for each parameter. Significant parameter estimates and odds ratios (OR) indicated in bold. Not significant (ns) OR omitted.**

| Parameter | ESS | Mean | SD | 2.5% | 97.5% | OR |
|---|---|---|---|---|---|---|
| Intercept | | | | | | |
| *Ski run class 1 (reference level)* | | 0 | | | | 1.00 |
| *Ski run class 2* | 2428 | **-3.68** | 0.78 | -5.25 | -2.17 | **0.03** |
| *Ski run class 4* | 2440 | **-2.46** | 0.78 | -4.00 | -0.96 | **0.09** |
| *Ski run class 5* | 2434 | **-3.13** | 0.76 | -4.64 | -1.68 | **0.04** |
| *Ski run class 6* | 2363 | **-4.70** | 0.75 | -6.18 | -3.25 | **0.01** |
| Relevant hazard rating | | | | | | |
| *Ski run class 1 (reference level)* | | 0 | | | | 1.00 |
| *Ski run class 2* | 2475 | **3.57** | 1.28 | 1.09 | 6.07 | **35.52** |
| *Ski run class 4* | 2336 | 0.74 | 1.22 | -1.60 | 3.12 | ns |
| *Ski run class 5* | 2368 | **-3.07** | 1.22 | -5.46 | -0.66 | **0.05** |
| *Ski run class 6* | 2435 | -2.24 | 1.25 | -4.71 | 0.15 | ns |
| Deep persistent slab | | | | | | |
| *Ski run class 1 (reference level)* | | 0 | | | | 1.00 |
| *Ski run class 2* | 3711 | 0.60 | 0.82 | -1.06 | 2.18 | ns |
| *Ski run class 4* | 2541 | -0.69 | 0.73 | -2.18 | 0.69 | ns |
| *Ski run class 5* | 2805 | **-2.33** | 0.75 | -3.86 | -0.94 | **0.10** |
| *Ski run class 6* | 3508 | **-2.66** | 0.82 | -4.33 | -1.17 | **0.07** |
| Persistent slab | | | | | | |
| *Ski run class 1 (reference level)* | | 0 | | | | 1.00 |
| *Ski run class 2* | 3283 | 0.36 | 0.52 | -0.64 | 1.37 | ns |
| *Ski run class 4* | 3111 | -0.43 | 0.48 | -1.38 | 0.5 | ns |
| *Ski run class 5* | 3035 | -0.82 | 0.47 | -1.74 | 0.09 | ns |
| *Ski run class 6* | 3106 | **-1.19** | 0.48 | -2.15 | -0.26 | **0.30** |
| Storm slab | | | | | | |
| *Ski run class 1 (reference level)* | | 0 | | | | 1.00 |
| *Ski run class 2* | 2954 | 0.38 | 0.52 | -0.64 | 1.39 | ns |
| *Ski run class 4* | 2533 | -0.25 | 0.47 | -1.19 | 0.69 | ns |
| *Ski run class 5* | 2453 | -0.44 | 0.47 | -1.37 | 0.5 | ns |
| *Ski run class 6* | 2504 | -0.56 | 0.47 | -1.5 | 0.36 | ns |
| Wind slab | | | | | | |
| *Ski run class 1 (reference level)* | | 0 | | | | 1.00 |
| *Ski run class 2* | 3026 | 0.25 | 0.56 | -0.86 | 1.35 | ns |
| *Ski run class 4* | 2708 | 0.17 | 0.52 | -0.84 | 1.20 | ns |
| *Ski run class 5* | 2676 | 0.41 | 0.51 | -0.58 | 1.39 | ns |
| *Ski run class 6* | 2637 | 0.4 | 0.51 | -0.6 | 1.38 | ns |
| Cornice | | | | | | |
| *Ski run class 1 (reference level)* | | 0 | | | | 1.00 |
| *Ski run class 2* | 10000 | 1.99 | 1.77 | -1.2 | 5.77 | ns |
| *Ski run class 4* | 4411 | -0.51 | 1.12 | -2.76 | 1.65 | ns |
| *Ski run class 5* | 4320 | -1.09 | 1.08 | -3.26 | 0.97 | ns |
| *Ski run class 6* | 4249 | -0.09 | 1.07 | -2.26 | 1.94 | ns |
| Loose wet avalanches | | | | | | |
| *Ski run class 1 (reference level)* | | 0 | | | | 1.00 |
| *Ski run class 2* | 3507 | -0.94 | 0.92 | -2.81 | 0.82 | ns |
| *Ski run class 4* | 3519 | -0.54 | 0.94 | -2.43 | 1.24 | ns |
| *Ski run class 5* | 3345 | **-1.77** | 0.90 | -3.61 | -0.08 | **0.17** |
| *Ski run class 6* | 3471 | -1.31 | 0.93 | -3.21 | 0.43 | ns |





| Parameter | ESS | Mean | SD | 2.5% | 97.5% | OR |
|---|---|---|---|---|---|---|
| Loose dry avalanches | | | | | | |
| *Ski run class 1 (reference level)* | | 0 | | | | 1.00 |
| *Ski run class 2* | 10000 | 0.83 | 2.20 | -3.32 | 5.20 | ns |
| *Ski run class 4* | 10000 | -2.00 | 2.10 | -6.18 | 2.14 | ns |
| *Ski run class 5** | - | - | - | - | - | |
| *Ski run class 6** | - | - | - | - | - | |
| Wet slab | | | | | | |
| *Ski run class 1 (reference level)* | | 0 | | | | 1.00 |
| *Ski run class 2* | 5640 | 0.26 | 0.80 | -1.32 | 1.81 | ns |
| *Ski run class 4* | 5361 | 1.46 | 0.79 | -0.10 | 2.99 | ns |
| *Ski run class 5* | 10000 | 0.96 | 1.10 | -1.28 | 3.02 | ns |
| *Ski run class 6* | 10000 | -0.93 | 2.00 | -5.21 | 2.55 | ns |
| Run code previous day | | | | | | |
| *Not skied in previous week (reference level)* | | 0 | | | | 1.00 |
| *Skied in previous week* | 10000 | -0.37 | 0.68 | -1.67 | 1.04 | ns |

\* There are no cases in the dataset, where Loose Dry Avalanche Problems were specified for ski runs in classes 5 or 6.

Combining the group-specific intercept, which represents the base tendency of each group to be open, and the group-specific slope estimate, which shows how strongly the run list codings of a group of runs are affected by increasing hazard, provides a more comprehensive picture. While the odds of runs being open decrease with increasing avalanche hazard ratings in all ski runs classes, the magnitude of the decrease varies substantially (Table 3).

**Table 3: Odds ratios of each ski run classes being open with increasing avalanche hazard relative to Low avalanche hazard.**

| | Ski run class | | | | |
|---|---|---|---|---|---|
| **Hazard** | **Class 1** | **Class 2** | **Class 4** | **Class 5** | **Class 6** |
| Low | 1.000 | 1.000 | 1.000 | 1.000 | 1.000 |
| Moderate | 0.020 | 0.049 | 0.024 | 0.009 | 0.011 |
| Considerable | 0.010 | 0.059 | 0.014 | 0.002 | 0.003 |
| High | 0.005 | 0.072 | 0.009 | <0.001 | 0.001 |
| Extreme | 0.001 | 0.049 | 0.003 | <0.001 | <0.001 |

The odds of ski runs in Class 1 being open decreases by 1000 times as avalanche hazard goes from *Low* to *Extreme*. In comparison, ski runs in Class 2 are only 20 times less likely to be open with the same increase in avalanche hazard. This means that despite the lower overall tendency of runs included in this class to be open, the run list ratings of the Class 2 runs are less affected by danger ratings. Since many of these ski runs are located at or below tree line, we suspect that the observed pattern reflects that many of these runs offer safe skiing options through trees, even when avalanche hazard is elevated. The alpine terrain classes are much more strongly affected by changes in danger ratings as evident by the large negative slope estimates. The odds of ski runs in Class 4 being open decrease by 300 times with increasing hazard from *Low* to *Extreme*. The odds of





ski runs in Classes 5 and 6 being open decrease even by more than 1000 times. These alpine ski runs are substantially steeper. Moreover, many of the ski runs or pickup locations can be affected by overhead hazard.

Examining the odds of runs in a specific class being open at a specific avalanche hazard relative to Class 1 (Table 4) highlights the relative importance of the various ski run classes at different hazard ratings. For instance, the odds of runs in Class 2 being

open relative to Class 1 increases with increasing avalanche hazard rating. This pattern emerges from the fact that the odds of being open decrease more quickly in Class 1 than in Class 2 (Table 4). A similar pattern can be observed between ski run Classes 4 and 5. Runs of Class 4 are approximately 12 times less likely to be open at *Low* hazard conditions than ski runs of Class 1. Similarly, ski runs in Class 5 are approximately 22 times less likely to be open at *Low* hazard conditions than Class 1. However, the ski runs of Class 5 are closed much more quickly as avalanche increases. The relative odds for ski runs in Class 4

being open are more than 5 times smaller for *Extreme* avalanche hazard, the relative odds for ski runs in Class 5 are 500 times smaller. Ski runs in Class 6 are more than 100 times less likely to be open with *Low* hazard and 1000 times with *Extreme* avalanche hazard.

**Table 4: Odds ratios of each ski run classes being open with increasing avalanche**
**hazard relative to ski run class 1.**

| | Ski run class | | | | |
|---|---|---|---|---|---|
| **Hazard** | **Class 1** | **Class 2** | **Class 4** | **Class 5** | **Class 6** |
| Low | 1.000 | 0.025 | 0.085 | 0.044 | 0.009 |
| Moderate | 1.000 | 0.062 | 0.103 | 0.020 | 0.005 |
| Considerable | 1.000 | 0.150 | 0.124 | 0.009 | 0.003 |
| High | 1.000 | 0.367 | 0.149 | 0.004 | 0.002 |
| Extreme | 1.000 | 0.896 | 0.179 | 0.002 | 0.001 |

As expected, our results confirm that the appropriateness of runs for guiding decreases with increasing hazard. However, they also highlight that the effect of avalanche hazard on run list codes depends heavily on the type of terrain that is being assessed. Gentle and frequently skied terrain in all elevation bands with no or only minor exposure to avalanches slopes is much less

affected by avalanche hazard. Severe alpine terrain with exposure to either multiple smaller or even large avalanche slopes on the ski runs or exposure to overhead hazard is much more affected by an increase in avalanche hazard. It is important to note that overhead hazard is not only relevant when it affects a skiing line, but also when the associated pickup locations are threatened.

### 3.2   Effect of avalanche problems and terrain type

Our results show that only certain avalanche problem types influence run list codes and that their effect differs among ski run classes. The presence of *Deep persistent slab avalanche problems* exhibits a negative effect on ski runs in Classes 5 and 6. This means that runs in severe alpine terrain are much less likely to be open during times when *Deep persistent slab avalanche problems* are a concern (OR=0.10 and OR=0.07, respectively, Table 1). A similar trend emerged for *Persistent slab avalanche*




*problems*, but only for ski runs of Class 6, which showed a significant decrease in the likelihood of being open (OR=0.30). The presence of *Wet slab avalanche problems*, however, exhibited a negative effect on the likelihood of runs being open on all ski run classes (main effect OR=0.21, Table 1). Finally, we observed a negative effect of *Wet loose avalanche problems* on the severe runs in Class 5 (OR=0.17).

Compared to the effect of avalanche hazard ratings, the influence different avalanche problem types is considerably smaller as indicated by the smaller parameter estimates. While hazard ratings reflect the severity of the avalanche hazard conditions in general and affect run codings more globally, avalanche problem types modulate this effect for the specific avalanche situation. For instance, whereas the presence of a widespread *Storm slab avalanche problem* affects the likelihood of ski runs being open equally across all ski run classes, the presence of a *Deep persistent slab avalanche problem* results in a higher likelihood of

ski runs with severe alpine terrain with generally steeper or larger avalanche slopes being closed. Similarly, our results only show a significant effect of *Wet loose avalanche problems* on run list coding of severe alpine terrain. While these avalanches are typically confined to surface layers and therefore often small, the can gain size and speed. As such, terrain with severe consequences (e.g., somebody caught in an avalanche being carried into obstacles or over cliffs) seems to be assessed more cautiously.

## 3.3 Effect of run code of the previous day and recent skiing on a run

Whether a run was open the previous day and whether it was skied within the previous seven days have both a significant influence on it being open on any given day (Table 1). Compared to a run that had neither been skied during the previous seven days not was it open the day before, being open the day before increases a run's odds of being open by 20 times. The effect of having recently skied the run is even larger, as it increases the odds of a run that was closed the day before to be open by 31

times (Table 1). This can also be seen from the modelled probability curves for different hazard levels and operational scenario in Figure 4. Panel (b) illustrates the model results for a scenario where runs were not open the day before but recently skied and panel (c) shows a scenario where runs were open the day before and recently skied. In both cases, the curves are shifted to the right compared to the base scenario where runs were neither open the day before nor recently skied. We were somewhat surprised, however, by the fact that the interaction between these two parameters did not turn out to be significant.

Our results illustrate the strong effect of the run list from the previous day as terrain choices evolve over the course of a season. Terrain choices in mechanized skiing operations are made in stages and are constantly adjusted based on the conditions on the day before incorporating the incremental daily changes (Israelson, 2015). Moreover, the strong effect of previous skiing supports the often-expressed importance by guides of experiencing the conditions and having recent first-hand field observations. This effect is even more important than being open the previous day. As the season progresses, runs that have

been skied before and where the guiding team has recent observations about the specific conditions on that run are opened more quickly than comparable runs where such recent experiences are lacking. Previous skiing is an important part of managing risk in heli-skiing as it is considered as a compaction and stabilization factor (Clair Israelson, personal communication, 2019).



Together, these effects underline the necessity for analysing professional terrain choices in their temporal context. While revealed terrain preference data from GPS tracking units (e.g., Hendrikx et al., 2016; Thumlert and Haegeli, 2018) offer promising avenues for learning about professional avalanche risk management expertise at spatial scales below the run level, it is important to remember that these terrain decisions cannot be analysed as independent, isolated samples as they are always made in an operational context. It is therefore imperative to analyse the observations in the proper temporal context (i.e., open previously, skied previously) and spatial context (run list codes, run use, skied line on a run) to extract meaningful relationships between hazard and terrain choices that can be generalized.

## 3.4    Random effects on run level

While random effects on the run level were highly significant in preliminary models that did not include ski run class as a covariate, they were mostly insignificant in our final model that included ski run class as covariate (Figure 5). This highlights that the ski run classes derived by Sterchi and Haegeli (2019) are able to capture the essence of the ski runs, and the realism of the results confirm the suitability of ski run characterization approach for analysing professional terrain choices in avalanche terrain in a quantitative way.

However, the random effects still provide useful insight into factors affecting run list choices of individual ski runs that are not captured by the fixed effects included in the model. Some ski runs are significantly less sensitive to avalanche hazard (shown in red, Figure 5), while others are coded significantly more sensitively with respect to an increase in avalanche hazard (shown in blue, Figure 5). The run "Sea of Cortez" (Class 4) is significantly less open than the rest of this group of ski runs when *Deep persistent slab avalanche problems* are of concern. We suspect that this difference might be caused by the fact that a more severely exposed line of this ski run can be affected by large overhead avalanche hazard. Similarly, the ski run "Pacha Mama" (Class 2) is significantly less open under conditions with higher hazard than the rest of the group. While the least severe ski line at treeline on this run only has minor exposure to avalanche hazard, more severe sections of the run are also exposed to overhead hazard. In both cases, we suspect that such a configuration might also affect the decision to close run sections that have no exposure to avalanche hazard. The ski run "Shrek" (Class 6) exhibits another interesting pattern. While it has a negative random intercept indicating that it is significantly less open than the rest of its class, it is significantly more open when *Deep persistent slab* or *Persistent slab avalanche problems* are a concern, or with increased avalanche hazard. A detailed look at the characteristics of "Shrek" provides some insight into the reason behind this pattern. "Shrek" offers moderately steep skiing through glades and snow forest with an open canopy. While skiers are only exposed to smaller avalanche slopes, the run contains tree well hazard and was characterized as unfriendly and not preferred by the guiding team. Based on this characterization, we suspect that "Shrek" is a unfavoured run that is generally closed but potentially opened when operationally needed (i.e., when challenging hazard conditions restrict other skiing options).

These examples highlight that certain individual attributes of ski runs can be responsible for significant deviations from the typical assessment of ski runs of similar terrain type.





**Figure 5: By-run random effects. The dots indicate the mean OR whereas the line represents the 95% credible interval. Blue and red dots indicate OR that are significantly smaller or larger than 1 (i.e., credible interval does not cross 1).**





## 3.5 Seasonal differences

We included a second random effect in the analysis to account for the particularities of individual seasons. The resulting random intercepts for season reflect differences in the general propensity of runs being open in each season (Figure 6). Our result show that runs were coded open less than half as often during the low snowpack winter of 2014 compared to other
5   seasons (OR = 0.3). Overall, winter 2014 was characterized by record low snowpack heights which especially affected the closure of low elevation ski runs due to the marginal snowpack or increased skiing hazards for the guests. In addition, a persistent weak layer that was buried mid-season and remained a concern for the remainder of the season was responsible for the closure of the more severe ski runs.

  This result highlights that having long-term datasets is critical for identifying meaningful patterns in risk management practices
10   as the unique characteristics of individual winters can affect observed choices considerably. Since we are interested in extracting generalizable terrain choice rules, it is important to work with a statistical method that is able account for such random deviations. Hence, mixed effects models are an excellent approach for analysing terrain choices as they properly account for the nested structure of terrain selection datasets.

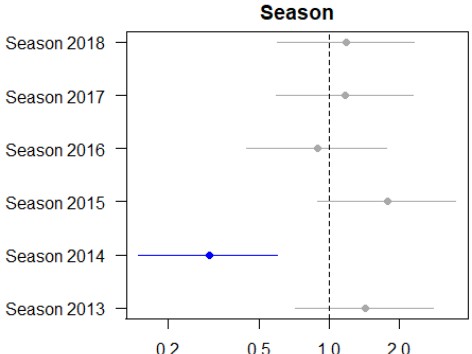

**Figure 6: By-season random effects. Same presentation as Figure 5.**

## 3.6 Limitations and future challenges

20   While the present results offer valuable quantitative insight into the relationship between avalanche hazard and run list codings at NEH, there are several potential avenues for exploring these relationships further and developing operational decision aids that offer value to guides team. While the present model only included a relatively crude representation of avalanche hazard (i.e., hazard rating and presence of avalanche problems), a more complete characterization of avalanche hazard according to the CMAH (Statham et al., 2018) could reveal more detailed insights about the suitability of runs under specific avalanche



hazard conditions. For example, explicitly including the likelihood of avalanches and destructive size parameters of the existing avalanche problems in the run list model has the potential to extract more detailed information about the relationship between the avalanche hazard situation and characteristics of runs with acceptable skiing terrain. Similarly, integrating more detailed ski run characteristics into the analysis might also help to reveal additional insight. Even though it was a conscious choice to

use an operation-specific ski run classification to represent the nature of the terrain in the present study to limit the complexity of this first quantitative analysis to a reasonable level, future research in this area will need to isolate the operation-specific intricacies so that the identified patterns between avalanche hazard and terrain that can be generalized across operations. However, taking this research to this level will require operational datasets of run list choices and avalanche hazard information that are substantially larger than the dataset used in the present study.

**4    Conclusions**

Using a large, multi-seasonal dataset of operational run list choices from a mechanized skiing operation, we applied a general linear mixed effects model to quantitatively explore the relationship between avalanche hazard conditions and acceptable skiing terrain numerically for the first time. Mixed effects models including random effects are an adequate statistical tool for analysing terrain choices since they can properly account for the nature of the dataset with its repeated measures (i.e., panel

structure). Our model included an avalanche hazard rating and eight binary variables indicating the presence of different avalanche problem types as predictors and the class of the ski run, whether it was skied in the previous seven days and how it was rated on the previous day as covariates. The model included by-run and by-season random effects.

Our results show that the effect of avalanche hazard on run list codes depends heavily on the type of terrain that is being assessed. While the run list ratings of the gentlest terrain are only marginally affected by hazard ratings, severe alpine terrain

is especially susceptible to increasing avalanche hazard. Compared to the effect of the avalanche hazard rating, the effects of the different avalanche problem types on the run list codes are small but represent critical, ski run class specific adjustments. Our results also highlight the strong effect of recent skiing and thus experiencing the conditions and having recent first-hand field observations on run list codings. This result reflects the fact that guides reopen runs they have recently skied more quickly than other comparable runs. Previous skiing is an important part of managing risk in heli-skiing as it is considered as a

compaction and stabilization factor (Clair Israelson, personal communication, 2019). The strong effect of the run code of the previous day highlights that terrain choices in mechanized skiing are evolving over the course of a season and further underline the necessity for analysing professional terrain choices in their temporal context.

While our results primarily confirm expectations, we believe this study provides a valuable step towards describing the terrain selection process at mechanized skiing operations numerically in a meaningful way. For the first time, the effect of avalanche

hazard has been isolated from the influence of other factors such as the run list code the day before and the effect of recent skiing. Properly isolating these effects is critical for describing the relationship between avalanche hazard and acceptable terrain in a meaningful fashion. In addition to offering insight into the run list coding process, the present research also provides


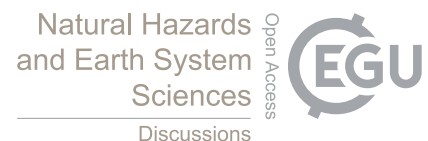

important context for the analysis of small-scale terrain choices in avalanche terrain (e.g., analysis of GPS tracks) since the terrain choices in mechanized skiing are made in stages and the decisions made in the field critically depend on the choices of eliminating unsuitable runs made during the preceding guide meeting.

In the long-term, this body of research will develop the foundation for the design of evidence-based operational decision aids that can help guides to make terrain choices more efficiently. It is important to note that we do not envision these decisions aid to actually make guiding decisions or be used for external auditing purposes like suggested by Hendrikx et al. (2016). However, if designed correctly, such decision aids may offer an independent reference that allows guides to check their morning run lists against their own historical decisions under similar conditions. Furthermore, the knowledge gained from these models may create the necessary foundation for the development of evidence-based terrain guidance tools for recreationists in the future.

## 5    Acknowledgments

We would like to thank Northern Escape Heli Skiing for their willingness to participate in this study. The NSERC Industrial Research Chair in Avalanche Risk Management at Simon Fraser University is financially supported by Canadian Pacific Railway, HeliCat Canada, Mike Wiegele Helicopter Skiing, the Canadian Avalanche Association. The research program receives additional support from Avalanche Canada and the Avalanche Canada Foundation. Reto Sterchi was also supported by SFU's Big Data Initiative KEY and a Mitacs Accelerate fellowship in partnership with HeliCat Canada. Northern Escape Heli Skiing is a member of HeliCat Canada.

## 6    Author contributions

RS, PH and PM co-led the design of this study. RS conducted the statistical analysis and authored the initial draft of the paper. PM consulted on the statistical analysis. RS, PH and PM subsequently edited the paper collaboratively to produce the final version of the paper.

## 7    Competing interests

The authors declare that they have no conflict of interest.

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
