# Peer review of "Exploring the relationship between avalanche hazard and large-scale terrain choices at a helicopter skiing operation - Insight from run list ratings"

_Natural Hazards and Earth System Sciences, 2019_

## Referee Comment (RC1) · Anonymous Referee #1 · 20 Apr 2019

The authors provide a study on the relationship between avalanche hazard conditions and ski terrain choices based on a general linear mixed effects model using data from a mechanised skiing operator in Canada. Based on an avalanche hazard rating and eight variables describing the type and severity of avalanche problem, as well as observed decisions on a set of ski runs originating from the commercial skiing operator, the authors show that the effect on hazard conditions on the run list codes (and therefore the question of whether or not paid operation can be undertaken on a specific run during a specific day) depends on the type of terrain being evaluated. Moreover, some

other insightful results are shown, such as the re-opening of runs based on recent (accident-free) experience of a guide having run the location before. The topic is on the interface between snow avalanche science and practice, suitable for the target journal and therefore, publication is recommended.

I only have some minor comments that may be addressed before acceptance:

- The tile uses the expression "large-scale"; I recommend the use of "regional" here so that it becomes clear that a large scale (1:10,000 or so) is meant, or "detailed assessment" if this should be the focus, but not – as this expression is quite often also used in NHESS – a nation-wide assessment.

- In the abstract as well as in the main text body the authors repeatedly address the term "acceptable risk level", from the overall scientific discussion and concept behind risk and vulnerability, I am wondering what exactly is meant by "acceptable" (death rates below a certain percentage? Number of ski runs without avalanche accident?) and if some explanatory sentences could help here to avoid confusion.

- The authors address multiple times the "mechanised skiing operation" but are using data from one operator; maybe the wording could be "mechanised skiing operator" to avoid confusion (e.g., page 1, line 11; page 20, line 11).

- On page 2, lines 1-22 the author describe the procedure of assessing avalanche hazard and establishing the run list, it would be useful to underpin this by a Figure showing the different steps by e.g., boxes and arrows in between.

- Please check references for updates, and provide a doi for those references that are in press.

---

## Referee Comment (RC2) · Anonymous Referee #2 · 13 May 2019

General comments: The relationship between avalanche hazard and skiing runs of an heliskiing operation was described quantitatively using a general linear mixed effects model. The results show that whether runs in gentle terrain or below treeline were skied hardly depends on the avalanche hazard, whereas runs in complex alpine terrain are affected by the avalanche hazard. The effect of avalanche problems is diverse. Some problems effect the closure of runs (e.g. deep persistent slab problem). Further the results show that runs which have recently been skied are open more quickly than others in comparable terrain. Although these results are not surprising, they confirm

habits of professional guides quantitatively which is new. The description of data and methods is mostly clear and understandable. The quality of text, tables and most figures are appropriate (details below). The paper is a valuable contribution confirming important factors for route selection in a quantitative way. The results can contribute for further development of tools to assist decision making on opening runs for mechanised skiing. The paper is acceptable with minor revisions.

Specific comments: Title and Abstract underline the content of the paper.

Methods: Page 4, line 15: "Overall, Statham et al.(2018) describe . . . . . . (delete "and") Page 4, lines 20 to 31: this content does not really belong to the description of the data. In my opinion it also could be skipped. Page 5, lines 12 to 30: This part rather belongs to the introduction and could be adapted in a way to emphasise the motivation for this study. Page 5, line 18: Better talk about avalanche sizes on figures 1-3 e.g. because the wording has changed in the European classification. Page8/9: The explanatory variables and interactions are well explained but could be summarized in a table for a better overview. Further the illustration and explanation of the model is not clear. Better describe model with a formula than with figure 3. Or change Fig.3 for better understanding. Page 10, lines 17 to 24: This section rather fits to the results chapter and explaines Fig. 4.

Results: Page 11, line 8: Mention value in the text (e.g. in brackets) for better understanding. Page 12, Fig. 4: Shading in graphs is not clear. What is 50%, 80% and 95%. Better reduce to 2 percentages. Label of x-axis is missing. Mention avalanche hazard as x-axis in caption text. Page 15, line 28: Table 2 not 1 Page 18, Fig. 5: Figure is to small and not readable. Label of x-axis is missing.

Figures: Fig. 2: Is rather small. Could be expanded to entire page width. Fig. 4: Label of x-axis is missing Fig. 5: Figure is to small and not readable. Label of x-axis is missing.

Technical corrections: Page 16, line 5: Typo: " . . . . . ., the influence of different. . . . . . . . ."
Page 16, line 12: Typo: " . . . . . .., they can gain size and speed." Page 19, line 4: Typo: "s" is missing either for "results" or "shows" Page 19, line 12: Typo: ". . . .. method that is able to account for . . ..." Page 21, line 5: Typo: " . . . . . ..envision these decision aids to . . . . . ."

---

## Referee Comment (RC3) · Anonymous Referee #3 · 21 May 2019

General Comments: The authors present a study of the effect of avalanche hazard on run list terrain decisions from a helicopter skiing operation in northern Canada. From my knowledge, the statistical model chosen is appropriate. Overall, these terrain selection decisions of a helicopter skiing guiding team are complex and this paper provides the first meaningful insight into many of the relevant factors. Excellent work. While run list decisions are limited in their actual usefulness to understand guide's management of avalanche risk, as accurately detailed in the paper, they are an important part of the hierarchical decision-making process. This paper will help build the foundation for

future studies.

The main results show many expected patterns, and I am not aware of any other studies which provide this quantitative evidence of guide's decision-making.

Publication is recommended with revisions and some clarification.

Statistical model: Note, while I do have experience using logistic regression models and overall they seem appropriate for this analysis, I do not have sufficient expertise with these models to provide a meaningful evaluation of the applied model used in this study.

Specific Comments:

Title: Consider "Exploring the relationship between avalanche hazard conditions and run-list terrain choices at a helicopter skiing operation". A little shorter and the phrase "large-scale terrain choices" is ambiguous.

Page 1 Line 24-25: Please check if there is a more recent reference. Walcher et al. (under review) or Walcher (Master thesis) perhaps would be more appropriate.

Page 1 Line 27 - 29: Consider mentioning that operations use direct control of avalanche hazard through the use of explosives and strategic control of future avalanche hazard through "run maintenance" skier traffic.

Page 2 Line 3-5: General comment for reference, most mechanized guiding teams will produce the avalanche hazard forecast for the first run or two of the day rather than for the full day. i.e. "what is the avalanche hazard as we head out the door?". This hazard evaluation is then updated as new information is obtained throughout the day.

Page 2 Line 3 - 5: Consider adding brief details about 'avalanche problems' as these are more impactful on the run list than the avalanche hazard rating.

Page 2 Line 7: Please change "...open or closed for skiing with guests..." to "...open or closed for guiding with guests...". Note, disregard this if the specific operation (Northern

Escape) uses the stated nomenclature.

Page 3 Figure 1: Please add direction indication (i.e. north arrow) and coordinates.

Page 4 Line 1 - 8: Does NEH use yellow coding of runs that can be opened in the field after a specific condition has been confirmed? If so, could you comment on how this might affect the results of the study?

Page 5 Line 4: Delete "for a given path". Avalanche size classification relative to the path size is the US relative scale size definitions and the Canadian size definitions are referenced.

Page 5 Line 8 - 10: Consider deleting the sentence "While this hazard ... ". This is not directly relevant to the study and can be discovered through the references.

Page 5 Line 12 - 28: Please consider deleting these lines and re-wording. The background information on avalanche terrain classification, while interesting, is not very relevant. In my opinion, it would be more beneficial to focus on the methods used in this study to encode the runs and the benefits of these methods. The Wakefield et al., 2018; and Sterchi and Haegeli, 2019; studies are appropriate to describe and to describe how they were applied here in this study.

Page 6 Line 19 - 20: Please re-word or delete "or non-glaciated or glaciated alpine".

Page 6 Figure 2: Increase the size of the Figure with the aim to increase the font size. It is difficult to read the run labels.

Page 6 Figure 2 caption: Change the word "average" to "boxplots" or something similar that describes what data are shown.

Page 6 Line 18 to Page 7 Line 14: Consider using a table to describe the characteristics of the 6 classes of runs. Consider example photographs of the terrain from each code as these would greatly enrich the understanding of the terrain types.

Page 7 Line 9: Consider re-wording "Life-changing".

Page 8 Line 27: Page 4 Line 15 details that the CMAH uses nine avalanche problems. It appears as though you removed glide-slab problem from the analysis, which seems appropriate, however could you provide the rational for this?

Page 8 Line 31 -32: The CMAH specifies "unlikely" as the lowest likelihood term, how were avalanche problems assessed lower than "unlikely"?

Page 8 Line 29 - 31: This sentence is not entirely accurate. Size 1 avalanches are "relatively harmless to people", whereas Size 1.5 avalanches are not specifically defined and are somewhere between Size 1 "relatively harmless to people" and Size 2 "could injure, bury or kill a person". Further, the analysis would likely be more insightful with avalanche problems assessed with Size 1.5 avalanches included. The avalanche problem "Loose Dry" is often associated with smaller more predictable avalanching and often isn't assigned avalanche sizes larger than 1.5. Saying that, better insights into the "Loose Dry" avalanche problem will not substantially alter the results of the paper, so I leave it to the authors to decide whether to change the analysis.

Page 12 Figure 4: Label the X-axis and increase font size for the axes.

Page 12 Figure 4 caption: Add details that the x-axis represents the relevant avalanche hazard rating.

Page 18 Line 23 - 30: Inspecting Figure 5 for the run "Shrek", I do not observe the negative random intercept: it appears to be non-significant and slightly positive. It does appear to show significant positive OR for Deep Persistent Slabs and Persistent Slabs. Please explain.

Page 18 Figure 5: -Please increase font sizes as this figure is nearly unreadable. - Change the x-axis for "Relevant Avalanche Hazard Rating" to match the other formats. -Overall, I might challenge the authors to consider if there would be another graphical format that might convey the key points of this Figure more clearly and concisely. - One of the fascinating results from this plot is the increased variance in OR between

avalanche problems, for example the OR for each run under Deep Persistent Slabs and Persistent Slabs show much higher variance compared to the more predictable avalanche problems like Storm Slabs / Dry Loose / Wet Loose. The Relevant Hazard Rating also shows higher relative variance in ORs. - A very insightful set of results that are likely available with this dataset and analysis would be the relative difference of run coding probabilities between avalanche problems with increasing levels of avalanche hazard ratings. i.e., Produce Figure 4 graphs for grouped avalanche problems (Storm and Wind and Loose Dry, Persistent and Deep Persistent, Wet Slab and Wet) or each individual problem, and remove the recency of skiing on the runs classification.

Technical Corrections:

Page 3 Figure 1 caption: Delete "Geographical". It is obvious that it is a map.

Page 4 Line 5: Change "(i.e., the run is safe to ski with guests)" to "(i.e., the run is available to guide with guests)".

Page 4 Line 15: Delete reference"(Statham et al., 2018)". The CMAH has already been referenced.

Page 5 Line 21: Reword "at the runs scale".

Page 5 Line 33: Add "(2019)" after Haegeli.

Page 6 Line 1: Change "are" to "were"

Page 6 Line 6: Delete "(Sterchi and Haegeli, 2019)". The study has already been referenced.

Page 6 Figure 2 caption: Please confirm whether the Sterchi and Haegeli study is under review or has been published 2019, then update the manuscript accordingly.

Page 16 Line 12: Typo.. "the" should be "they".
* * *
2019-57, 2019.

---

## Author Comment (AC2) · 6 Jul 2019

**Exploring the relationship between avalanche hazard and large-scale terrain choices at a helicopter skiing operation – Insight from run list ratings**

**Response to Anonymous Referee #2**

Reto Sterchi, Pascal Haegeli
July 6, 2019

We would like to thank the reviewer for taking the time to read our manuscript in detail and provide constructive feedback. The following sections describe our response to the comments raised by the referee and outline the changes we made to the manuscript to address these concerns.

**1    Methods: Description of avalanche problems with examples**

**Review**
*[…] Page 4, lines 20 to 31: this content does not really belong to the description of the data. In my opinion it also could be skipped. […]*

**Response to the review and changes made to the manuscript**
The avalanche problem types are a crucial part of the Conceptual Model of Avalanche Hazard (CMAH) and the data set used for our study. However, we agree that a brief description of the importance of identifying avalanche problems and their connection to terrain choices might be enough information so that readers can understand what we did in our study and the essence of the results but can refer them to Statham et al (2018) for the details. We shortened and changed the text of lines 20-31 as following.

*[…] "While some avalanche problems are of relatively short duration and can be managed easily by avoiding specific terrain features within runs (e.g., wind-loaded slopes when a wind slab avalanche problem is present), others can persist for weeks, even months and require a more conservative risk management approach that includes a broader range of terrain (Haegeli et al., 2010; Statham et al., 2018)." […]*

**2    Methods: Encoding the nature of the ski terrain**

**Review**
*[…] Page 5, lines 12 to 30: This part rather belongs to the introduction and could be adapted in a way to emphasise the motivation for this study.*

**Response to the review and changes made to the manuscript**
A similar comment was made by reviewer #3. We shortened and changed the text of lines 11-30 as following.

Page 5, line 11ff
*[…] To identify meaningful patterns between avalanche hazard and terrain choices numerically, it is critical to encode the nature of the available ski runs in a concise, but insightful way. To comprehensively capture of complex nature of entire ski runs into our model in a way that reflects how professional guides perceive them, we used the approach introduced by Sterchi and Haegeli (2019), which groups the ski*

*runs into operation-specific terrain classes based on multi-seasonal patterns in run list ratings (i.e., revealed terrain preferences). In comparison to existing terrain classification systems with small numbers of universal terrain classes (e.g., ATES; Statham et al., 2006; Campbell and Gould, 2013), Sterchi and Haegeli's approach identifies high-resolution, operation-specific ski run hierarchies based on multi-seasonal patterns in run list ratings (i.e., revealed terrain preferences). Sterchi and Haegeli first identified groups of ski runs by clustering similarly coded ski runs over the course of several winter seasons. Subsequently, they arranged the identified groups into a hierarchy that ranges from runs that are almost always open to runs that are only open when conditions are favourable. To better understand the nature of the revealed ski run classes, the authors had a senior lead guide at each participating operation provide a comprehensive but structured description of their ski runs with respect to access, type of terrain, skiing experience, operational role, hazard potential, and guide-ability. Since this ski run classification is based on past operational risk management decisions, it reflects the local terrain expertise and avalanche risk management practices in the context of the available terrain and local snow and avalanche climate conditions (Sterchi and Haegeli, 2019). Thus, this approach represents a more meaningful characterization of ski run classes to analyze professional terrain choices in mechanized skiing operations. […]*

**3    Methods: Avalanche sizes**

**Review**
*[…] Page 5, line 18: Better talk about avalanche sizes on figures 1-3 e.g. because the wording has changed in the European classification.*

**Response to the review**
Thanks for highlighting this inconsistency in avalanche size description.

**Changes made to the manuscript**
To address the reviewer's concern, we made the following changes (highlighted in green):

*[…] and the potential of being seriously injured or deeply buried by avalanches of smaller or equal to size 3. […]*

**4    Methods: Model description**

**Review**
*[…] Page 8/9: The explanatory variables and interactions are well explained but could be summarized in a table for a better overview. Further the illustration and explanation of the model is not clear. Better describe model with a formula than with figure 3. Or change Fig.3 for better understanding.*

**Response to the review and changes made to the manuscript**
Thank you for pointing that out. After considerable reflection, we believe that a formula would not provide much clarification of the model due to the many variables and interactions involved. However, we believe that structuring the figure in a more table-like layout with additional variable information on could help to overcome the highlighted shortcomings. To address the reviewer's concern, we made the following changes to the figure.

[Figure]

**5 Methods: Description of result presentation**

**Review**
*[…] Page 10, lines 17 to 24: This section rather fits to the results chapter and explains Fig. 4. […]*

**Response to the review and changes made to the manuscript**
We agree that this description of the graph can also be moved into the results section and moved it into section 3.1 where we present figure 4.

**6 Results: Description of parameter estimate**

**Review**
*[…] Page 11, line 8: Mention value in the text (e.g. in brackets) for better understanding. […]*

**Response to the review and changes made to the manuscript**
Thanks for pointing out this inconsistency. To address the reviewer's concern, we added the parameter estimates on several instances throughout the results section.

**7 Results: Falsely referenced table**

**Review**
*[…] Page 15, line 28: Table 2 not 1 […]*

**Response to the review**
Thanks for highlighting this typo. We made the following changes (highlighted in green):

Page 15, line 28
*[…] This means that runs in severe alpine terrain are much less likely to be open during times when Deep persistent slab avalanche problems are a concern (OR=0.10 and OR=0.07, respectively, Table 2) […]*

**8    Figures: Size of figure 2**

**Review**

*[…] Fig. 2: Is rather small. Could be expanded to entire page width.  […]*

**Response to the review and changes made to the manuscript**

A similar comment was made by reviewer #3. We agree with the reviewers and propose to increase the size of the figure and will use the entire width of the page for the figure.

[Figure]

**9    Figures: Figure 4**

**Review**

*[…] Shading in graphs is not clear. What is 50%, 80% and 95%. Better reduce to 2 percentages. Label of x-axis is missing. Mention avalanche hazard as x-axis in caption text.  […]*

**Response to the review and changes made to the manuscript**

Thank you for pointing out this shortcoming of Figure 4. We agree that the including three different percentages is too much and makes the different shadings difficult to distinguish. To address the reviewer's concern, we will only use two percentages (50% and 95%). We also addressed the missing label of the x-axis and mentioned the axis in the caption text (highlighted in green).

[Figure]

*[…] Figure 4: Probabilities of ski runs being open for Storm slab avalanche problems shown for increasing hazard levels with (a) a scenario where ski runs were neither open previously nor skied recently, (b) a scenario where runs were not open the day before but recently skied, and (c) a scenario where runs were open the day before and recently skied. The visualizations include probability intervals of 50% and 95% for each ski run class as a whole based on 50 draws from the posterior distribution. Average daily percentages of open runs per ski run class are plotted as points where observations for this scenario exist in the dataset. […]*

**10   Figures: Figure 5**

**Review**
*[…] Fig. 5: Figure is to small and not readable. Label of x-axis is missing. […]*

**Response to the review and changes made to the manuscript**
Thank you for point this out. We replaced this figure in response to a comment of reviewer #3.

**11   Technical corrections**

**Review**
*[…] Page 16, line 5: Typo: "…, the influence of different …"[…]*

**Response to the review and changes made to the manuscript**

Thank you for point this out. We changed the sentence accordingly.

**Review**

*[…]* Page 16, line 12: Typo: "…, they can gain size and speed." *[…]*

**Response to the review and changes made to the manuscript**

Thank you for point this out. We changed the sentence accordingly.

**Review**

*[…]* Page 19, line 4: Typo: "s" is missing either for "results" or "shows" *[…]*

**Response to the review and changes made to the manuscript**

Thank you for point this out. We changed the sentence accordingly ("results").

**Review**

*[…]* Page 19, line 12: Typo: "… method that is able to account for …" *[…]*

**Response to the review and changes made to the manuscript**

Thank you for point this out. We changed the sentence accordingly.

**Review**

*[…]* Page 21, line 5: Typo: "… envision these decision aids to …" *[…]*

**Response to the review and changes made to the manuscript**

Thank you for point this out. We changed the sentence accordingly.

---

## Author Comment (AC3) · 6 Jul 2019

**Exploring the relationship between avalanche hazard and large-scale terrain choices at a helicopter skiing operation – Insight from run list ratings**

**Response to Anonymous Referee #3**

Reto Sterchi, Pascal Haegeli
July 6, 2019

We would like to thank the reviewer for taking the time to read our manuscript in detail and provide constructive feedback. The following sections describe our response to the comments raised by the referee and outline the changes we made to the manuscript to address these concerns.

**1    Title**

**Review** (Reviewer #1 made a related comment)
*[…] Consider "Exploring the relationship between avalanche hazard conditions and run-list terrain choices at a helicopter skiing operation". A little shorter and the phrase "large-scale terrain choices" is ambiguous. […]*

**Response to the review and changes made to the manuscript**
Thank you for this comment. Reviewer #1 had similar concerns about the term "large-scale". We think the proposed modification is an excellent suggestion that makes the title clearer.

Adapted title: "*Exploring the relationship between avalanche hazard conditions and run-list terrain choices at a helicopter skiing operation*"

**2    Reference: Walcher et al.**

**Review** (Reviewer #1 made a related comment)
*[…] Page 1 Line 24-25: Please check if there is a more recent reference. Walcher et al. (under review) or Walcher (Master thesis) perhaps would be more appropriate […]*

**Response to the review and changes made to the manuscript**
Thank you for highlighting this. The paper is in press and should have a DOI shortly.

*Walcher, M., Haegeli, P., and Fuchs, S.: Risk of death and major injury from natural winter hazards in mechanized backcountry skiing in Canada, Wild. Environ. Med., in press.*

**3    Introduction: Additional methods of controlling avalanche hazard**

**Review**
*[…] Page 1 Line 27 - 29: Consider mentioning that operations use direct control of avalanche hazard through the use of explosives and strategic control of future avalanche hazard through "run maintenance" skier traffic. […]*

**Response to the review and changes made to the manuscript**
Thank you for pointing this out. We agree with the reviewer that it is worth mentioning that depending on the operational practices, the use of explosives or the strategic control of the snowpack through skier

traffic is common. To address the reviewer's comment, we made the following changes (highlighted in green):

*Page 1, Line 27ff*
*[…] Operations manage this risk by continuously assessing the local avalanche hazard conditions and carefully choosing appropriate terrain and travel procedures to limit their exposure to avalanche hazard and keep the residual risk at an acceptable level while still providing a high-quality skiing experience.* *Some operations may use explosives to directly control avalanche hazard or purposely ski individual ski runs to control future avalanche hazard by modifying the local snowpack (often referred to as "run maintenance").* *[…]*

**4 Introduction: Hazard forecast only for first couple runs**

**Review**
*[…] Page 2 Line 3-5: General comment for reference, most mechanized guiding teams will produce the avalanche hazard forecast for the first run or two of the day rather than for the full day. i.e. "what is the avalanche hazard as we head out the door?". This hazard evaluation is then updated as new information is obtained throughout the day. […]*

**Response to the review and changes made to the manuscript**
Thank you for commenting that this sentence needs clarification. We fully agree with the comment of the reviewer and are proposing the following changes (highlighted in green) to better highlight the evolutionary character of the hazard assessment and run selection.

*Page 2, Line 3ff*
*[…] The daily process starts with a morning meeting where the guiding team assesses the current hazard conditions and produces a* *first* *large-scale avalanche hazard forecast across the entire tenure* *based on the previous day's experiences and the observed overnight changes.* *This* *initial* *hazard assessment is the foundation for the* *day's* *"run list", which represents the first terrain elimination filter. In this step, the guiding team goes through their inventory of predefined ski runs and collectively decides for each run whether it is open or closed for skiing with guests under the expected avalanche hazard conditions. It is important to note that depending on the nature of the operation, the scale of ski runs can range from tightly defined ski lines to areas the size of a medium ski resort. However, regardless of their size, the nature of ski run is consistent enough that they represent meaningful decision units at this stage of the risk management process. The large-scale, consensus-based run list that emerges from the morning meeting sets the stage for the skiing program of the day. Over the course of a skiing day,* *the avalanche hazard assessment* *is refined and adapted in response to direct field observations* *and runs that are skied are chosen from the run list accordingly.* *[…]*

**5 Introduction: Description of hazard assessment**

**Review** (Reviewer #2 made a related comment)
*[…] Page 2 Line 3 - 5: Consider adding brief details about 'avalanche problems' as these are more impactful on the run list than the avalanche hazard rating. […]*

**Response to the review and changes made to the manuscript**
We intentionally speak of avalanche hazard in general here in the introduction while we go into the details of how avalanche hazard is characterized with avalanche problems and an avalanche rating in the

methods section where we describe our data set. We did not make changes to the manuscript in response to this comment. However, please note that we revised the description of the avalanche problems included in our data set in response to Reviewer #2 (comment 1, manuscript page 4, lines 21ff).

**6    Introduction: General wording of run list codes**

**Review**
*[…] Page 2 Line 7: Please change "...open or closed for skiing with guests..." to "...open or closed for guiding with guests...". Note, disregard this if the specific operation (Northern Escape) uses the stated nomenclature. […]*

**Response to the review and changes made to the manuscript**
Thanks you for pointing out this inconsistency in the description of the codes in the methods section. To address the reviewer's comment, we made the following changes (highlighted in green):

*[…] In this step, the guiding team goes through their inventory of predefined ski runs and collectively decides for each run whether it is open or closed for guiding with guests under the expected avalanche hazard conditions. […]*

**7    Figure 1: North arrow and coordinates**

**Review**
*[…] Page 3 Figure 1: Please add direction indication (i.e. north arrow) and coordinates. […]*

**Response to the review and changes made to the manuscript**
Thank you for pointing out this cartographic flaw. To address the reviewer's comment, we changed the figure accordingly.

[Figure]

**8 Data set: Use of yellow coding of runs**

**Review**

*[…] Page 4 Line 1 - 8: Does NEH use yellow coding of runs that can be opened in the field after a specific condition has been confirmed? If so, could you comment on how this might affect the results of the study? […]*

**Response to the review and changes made to the manuscript**

Northern Escape does not use yellow codes to indicate that runs could be opened in the field conditional on specific conditions observed. No changes were made to the manuscript.

**9 Data set: Avalanche size classification**

**Review**

*[…] Page 5 Line 4: Delete "for a given path". Avalanche size classification relative to the path size is the US relative scale size definitions and the Canadian size definitions are referenced […]*

**Response to the review and changes made to the manuscript**

We agree with the comment and deleted "for a given path".

Page 5, line 3ff:

*[…] Destructive size is assessed according to the Canadian avalanche size classification (Canadian Avalanche Association, 2014) on a scale ranging from 1.0 (relatively harmless for people) to 5.0 (largest snow avalanche known , which 5 could destroy a village or a large forest area of approximately 40 hectares). […]*

**10 Data set: Description of avalanche hazard levels**

**Review**

*[…] Page 5 Line 8 - 10: Consider deleting the sentence "While this hazard … ". This is not directly relevant to the study and can be discovered through the references. […]*

**Response to the review and changes made to the manuscript**

We agree with the reviewer that this specification can be omitted from the description of the hazard rating applied by NEH. We made the following changes to the manuscript (additions and deletions highlighted in green and red respectively) to address this comment.

Page 5, line 7-10:

*[…] The hazard assessments for each elevation band are concluded by summarizing the overall hazard level that emerges from the combined avalanche problems with a single hazard rating on an ordinal scale from 1 (least hazardous) to 5 (most hazardous; Canadian Avalanche Association, 2015). […]*

New reference for the hazard rating:
Canadian Avalanche Association: Avalanche Hazard Rating Scale. InfoEx Advisory Committee. Available at http://infoexhelp.avalancheassociation.ca/wiki/Hazard_rating_definition_table (last access: 3 July 2019), 2015.

**11   Data set: Description of terrain classification**

**Review**

*[…] Page 5 Line 12 - 28: Please consider deleting these lines and re-wording. The background information on avalanche terrain classification, while interesting, is not very relevant. In my opinion, it would be more beneficial to focus on the methods used in this study to encode the runs and the benefits of these methods. The Wakefield et al., 2018; and Sterchi and Haegeli, 2019; studies are appropriate to describe and to describe how they were applied here in this study. […]*

**Response to the review and changes made to the manuscript**

A similar comment was made by Reviewer #2. We substantially shortened and changed the text of lines 11-30 as following.

Page 5, line 11ff

*[…] To identify meaningful patterns between avalanche hazard and terrain choices numerically, it is critical to encode the nature of the available ski runs in a concise, but insightful way. To comprehensively capture of complex nature of entire ski runs into our model in a way that reflects how professional guides perceive them, we used the approach introduced by Sterchi and Haegeli (2019), which groups the ski runs into operation-specific terrain classes based on multi-seasonal patterns in run list ratings (i.e., revealed terrain preferences). In comparison to existing terrain classification systems with small numbers of universal terrain classes (e.g., ATES; Statham et al., 2006; Campbell and Gould, 2013), Sterchi and Haegeli's approach identifies high-resolution, operation-specific ski run hierarchies based on multi-seasonal patterns in run list ratings (i.e., revealed terrain preferences). Sterchi and Haegeli first identified groups of ski runs by clustering similarly coded ski runs over the course of several winter seasons. Subsequently, they arranged the identified groups into a hierarchy that ranges from runs that are almost always open to runs that are only open when conditions are favourable. To better understand the nature of the revealed ski run classes, the authors had a senior lead guide at each participating operation provide a comprehensive but structured description of their ski runs with respect to access, type of terrain, skiing experience, operational role, hazard potential, and guide-ability. Since this ski run classification is based on past operational risk management decisions, it reflects the local terrain expertise and avalanche risk management practices in the context of the available terrain and local snow and avalanche climate conditions (Sterchi and Haegeli, 2019). Thus, this approach represents a more meaningful characterization of ski run classes to analyze professional terrain choices in mechanized skiing operations. […]*

**12   Data set: Terrain descriptors**

**Review**

*[…] Page 6 Line 19 - 20: Please re-word or delete "or non-glaciated or glaciated alpine". […]*

**Response to the review and changes made to the manuscript**

We simplified this sentence (changes highlighted in red):

Page 6, Line 19f:

[…] Most of the skiing is through open slopes at tree line, open canopy snow forest below tree line,  non-glaciated or glaciated alpine. […]

**13  Figure 2: Size and caption**

**Review**

*[…] Page 6 Figure 2: Increase the size of the Figure with the aim to increase the font size. It is difficult to read the run labels. […]*

*[…] Page 6 Figure 2 caption: Change the word "average" to "boxplots" or something similar that describes what data are shown. […]*

**Response to the review and changes made to the manuscript**
A similar comment was made by Reviewer #2. In response we increased the size of this figure to enhance readability. We also changed the caption of the figure (changes highlighted in green).

[Figure]

*Caption:*

*Figure 2: Boxplot of average seasonal percentages of run code 'open' for the 57 ski runs during the six seasons 2012/13 to 2017/18 with the six identified classes of similarly managed ski runs (Sterchi & Haegeli, 2019). Due to the small group size and their outlier characteristics, the two runs of Class 3 were not included in the present analysis.*

**14  Data set: Table with terrain characteristics**

**Review**

*[…] Page 6 Line 18 to Page 7 Line 14: Consider using a table to describe the characteristics of the 6 classes of runs. Consider example photographs of the terrain from each code as these would greatly enrich the understanding of the terrain types. […]*

**Response to the review and changes made to the manuscript**
We believe that providing photos of typical runs for each group add value to the presentation of the terrain characteristics and we added the following table to the manuscript:

Page 6, line 10:
*[…] Table 1 provides an overview of the general character of the NEH ski runs included in this study. […]*

*Caption for Table 1: Photos of typical ski runs included in this study. All photos reproduced with permission of NEH.*

| Class | Number of runs | Typical ski runs |
|-------|----------------|------------------|
| Class 1 | 8 | |
| Class 2 | 9 | |
| Class 3 | 3 | |
| Class 4 | 13 | |
| Class 5 | 12 | |

[Figure]

Class 6        13
[Figure]

[Figure]

**15    Data set: Descriptor "life-changing"**

**Review**

*[…] Page 7 Line 9: Consider re-wording "Life-changing". […]*

**Response to the review and changes made to the manuscript**

The description of the terrain classes is based on the study by Sterchi and Haegeli (2019). They used a survey that was developed in collaboration with senior lead guides to characterize and describe different terrain types. Since the descriptor "life-changing" originates form this survey, we did not make any changes to this manuscript.

**16    Data set: Number of avalanche problems**

**Review**

*[…] Page 8 Line 27: Page 4 Line 15 details that the CMAH uses nine avalanche problems. It appears as though you removed glide-slab problem from the analysis, which seems appropriate, however could you provide the rational for this? […]*

**Response to the review and changes made to the manuscript**

Thank you for commenting in this inconsistency. Since NEH does not specify glide slab avalanches, we only have eight avalanche problems in our dataset. We propose the following amendments for the manuscript (highlighted in green):

Page 4, Line 15ff

*[…] Overall, Statham et al. (2018) and describe nine[1] distinct types of avalanches problems (Dry loose avalanche problem, Wet loose avalanche problem, Storm slab avalanche problem, Wind slab avalanche problem, Persistent slab avalanche problem, Deep persistent slab avalanche problem, Wet slab avalanche problem, Glide avalanche problem, and Cornice avalanche problem) that differ in their development, avalanche activity patterns, how they are best recognized and assessed in the field, and what risk management strategies are most effective for managing them. […]*

*Footnote 1: Please note that NEH only uses eight types of avalanche problems as they do not specify Glide avalanches problems.*

**17    Data set: Avalanche problem likelihood**

**Review**

*[…] Page 8 Line 31 -32: The CMAH specifies "unlikely" as the lowest likelihood term, how were avalanche problems assessed lower than "unlikely"? […]*

**Response to the review and changes made to the manuscript**

Thank you for highlighting this issue. We realize that our description of the avalanche problem cases that were not included in the analysis was not clear in the original version of the manuscript. We considered cases were both the maximum and the typical likelihood of avalanches were both considered to be "unlikely" to be outliers and excluded them from the analysis. We changed the manuscript in the following way:

*[…] Because of the small number of cases, we also excluded avalanche problems where both typical and maximum likelihood were assessed*  *as "unlikely". […]*

**18  Data set: Exclusion of data point based on avalanche size**

**Review**

*[…] Page 8 Line 29 - 31: This sentence is not entirely accurate. Size 1 avalanches are "relatively harmless to people", whereas Size 1.5 avalanches are not specifically defined and are somewhere between Size 1 "relatively harmless to people" and Size 2 "could injure, bury or kill a person". Further, the analysis would likely be more insightful with avalanche problems assessed with Size 1.5 avalanches included. The avalanche problem "Loose Dry" is often associated with smaller more predictable avalanching and often isn't assigned avalanche sizes larger than 1.5. Saying that, better insights into the "Loose Dry" avalanche problem will not substantially alter the results of the paper, so I leave it to the authors to decide whether to change the analysis. […]*

**Response to the review and changes made to the manuscript**

We rerun the analysis as suggested and revised the content of the results section accordingly. The model calculations are robust, and all parameter estimates only differed in the sub-decimal range.

*Page 8, Line 29ff*

*[…] Since avalanches of Size 1.0*  *are considered relatively harmless to people (McClung and Schaerer, 2006), we only included avalanche problems in our analysis that were characterized with a maximum destructive size of at least Size 1.5. Because of the small number of cases, we also excluded avalanche problems where both typical and maximum likelihood were assessed*  *as "unlikely". […]*

**19  Figure 4: Readability, axis label and caption**

**Review**

*[…] Page 12 Figure 4: Label the X-axis and increase font size for the axes.*
*Page 12 Figure 4 caption: Add details that the x-axis represents the relevant avalanche hazard rating. […]*

**Response to the review and changes made to the manuscript**

Thank you for pointing out those graphical flaws. We adapted both the figure and the caption accordingly.

[Figure]

(a) Runs not open the day before and not skied recently

(b) Runs not open the day before but skied recently

(c) Runs open the day before and skied recently

*[…] Figure 4: Probabilities of ski runs being open for Storm slab avalanche problems shown for increasing hazard levels with (a) a scenario where ski runs were neither open previously nor skied recently, (b) a scenario where runs were not open the day before but recently skied, and (c) a scenario where runs were open the day before and recently skied. The visualizations include probability intervals of 50% and 95% for each ski run class as a whole based on 50 draws from the posterior distribution. Average daily percentages of open runs per ski run class are plotted as points where observations for this scenario exist in the dataset. […]*

**20 Figure 5: Observation for run "Shrek"**

**Review**

*[…] Page 18 Line 23 - 30: Inspecting Figure 5 for the run "Shrek", I do not observe the negative random intercept: it appears to be non-significant and slightly positive. It does appear to show significant positive OR for Deep Persistent Slabs and Persistent Slabs. Please explain. […]*

**Response to the review and changes made to the manuscript**

Thanks you for pointing out this inconsistency. This was a leftover from a previous draft that we forgot to adjust. We deleted the corresponding part of the results description accordingly.

**21 Figure 5: Graphical representation of results**

**Review**

*[…] Page 18 Figure 5: -Please increase font sizes as this figure is nearly unreadable. - Change the x-axis*

*for "Relevant Avalanche Hazard Rating" to match the other formats. - Overall, I might challenge the authors to consider if there would be another graphical format that might convey the key points of this Figure more clearly and concisely. - One of the fascinating results from this plot is the increased variance in OR between avalanche problems, for example the OR for each run under Deep Persistent Slabs and Persistent Slabs show much higher variance compared to the more predictable avalanche problems like Storm Slabs / Dry Loose / Wet Loose. The Relevant Hazard Rating also shows higher relative variance in ORs. - A very insightful set of results that are likely available with this dataset and analysis would be the relative difference of run coding probabilities between avalanche problems with increasing levels of avalanche hazard ratings. i.e., Produce Figure 4 graphs for grouped avalanche problems (Storm and Wind and Loose Dry, Persistent and Deep Persistent, Wet Slab and Wet) or each individual problem, and remove the recency of skiing on the run classification. [...]*

**Response to the review and changes made to the manuscript**

Thank you for pointing out this issue and the input into the variance of the by-run random effects for different avalanche problem types. We believe that this angle provides some valuable insight into our results and we therefore made the following changes to the manuscript:

- Presenting the random effects in a new table that shows their variance and lists ski runs with significant random effects as a foundation for the discussion in section "Random effects on run level (currently 3.4). We believe this presentation makes the results more insightful and we omit Figure 5.
- Discussing the overall insight from this with an additional subsection before discussing the effects of run code of the previous day and recent skiing on a run

*Table 5: Variance in by-run random effects expressed with the standard deviation per parameter. In addition, ski runs with significant positive or negative random effects are listed. The number in brackets indicate the ski run class.*

| Parameter | SD | Ski runs with significant random effects | |
|---|---|---|---|
| | | Positive random effect | Negative random effect |
| Intercept | 0.63 | Poison Beauty (5) | Donkey (4), Line King (5) |
| Relevant hazard rating | 1.12 | East Ridge (2), Back Door (5) | Pacha Mama (2), Tea Cup (2) |
| Deep persistent slab | 0.47 | Shrek (6) | Sea of Cortez (4) |
| Persistent slab | 0.23 | Back Door (5) | - |
| Storm slab | 0.06 | - | - |
| Wind slab | 0.06 | - | - |
| Cornice | 0.05 | - | - |
| Loose wet avalanche | 0.12 | - | - |
| Loose dry avalanche | 0.17 | - | - |
| Wet slab | 0.31 | - | - |

*3.3. Overall insight into the effect of avalanche hazard*

*Together, the main effects, interaction effects by ski run class and by-run random effects provide comprehensive insight into the overall effect of avalanche hazard (i.e., rating and avalanche problem*

*presence) on run list choices. While a significant main effect indicates that there is a consistent general response to changes in hazard across the entire run list, significant interaction effects mean that specific ski run groups respond differently from the overall pattern described by the main effect. Finally, significant by-run random effects show that individual runs substantially deviate from the general and/or ski run group specific response pattern.*

*The results of our analysis reveal that the run list ratings respond to the different aspects of avalanche hazard in different ways. The response to the hazard rating is characterized by a significant main effect (Table 1), significant interaction effects for some of the ski run classes (Table 2), and large variations in the by-run random effects with some of them being significant (Table 5). This means the observed general effect is superimposed with ski run group and ski run specific responses. The different avalanche problem types influence the run list ratings as follows. For Wet slab avalanche problems, only the main effect is significant (Table 1) indicating that all ski run classes respond to this avalanche problem the same way (Table 2). For Deep persistent avalanche problems and Persistent avalanche problems only certain ski run classes respond (i.e., no main effect, but ski run class specific interactions, Table 2), but certain individual ski runs significantly deviate from the overall class pattern with more variation in the by-run random effects (Table 5). For Loose wet avalanche problems, our model shows a non-significant main effect, some significant interactions effects for the different ski run classes and non-significant by-run random effects without any significant variability among runs. Finally, our model indicates no effect at all for Storm slab, Wind slabs, Cornices and Loose dry avalanche problems. This means that the response of the run list to these avalanche problem types is fully captured by the effect of the avalanche hazard rating.*

*Overall, the observed patterns in run list responses seem to be consistent with the existing understanding of different avalanche problems and the complexity of their management (Haegeli, Atkins and Klassen, 2010; Wagner and Hardesty, 2014). While the response to the simpler Storm slab, Wind slab, or Loose dry avalanche problems is fully captured by response to the hazard rating alone, the more complex Wet slab, Persistent slab and Deep persistent slab avalanche problems require more nuanced, avalanche problem specific terrain choices.*

*References:*

Haegeli, P., Atkins, R., and Klassen, K.: Decision making in avalanche terrain - a field book for winter backcountry users. Canadian Avalanche Centre, Revelstoke, BC, Canada, 2010.

Wagner, W. and Hardesty, D: Travel advice for the avalanche problems: A public forecasting tool. In: Proceedings of the International Snow Science Workshop, Banff, AB, Canada, 2014.

**22 Technical corrections**

**Review**
*[…] Page 3 Figure 1 caption: Delete "Geographical". It is obvious that it is a map.*

**Response to the review and changes made to the manuscript**
We deleted "Geographical" from the sentence.

**Review**

*[…] Page 4 Line 5: Change "(i.e., the run is safe to ski with guests)" to "(i.e., the run is available to guide with guests)".*

**Response to the review and changes made to the manuscript**

This is a valuable comment and we changed the sentence accordingly.

**Review**

*[…] Page 4 Line 15: Delete reference"(Statham et al., 2018)". The CMAH has already been referenced.*

**Response to the review and changes made to the manuscript**

This is a valid comment and we changed the sentence accordingly.

**Review**

*[…] Page 5 Line 21: Reword "at the runs scale".[…]*

**Response to the review and changes made to the manuscript**

Thanks you for pointing this out. We changed the sentence to "at the run scale".

**Review**

*[…] Page 5 Line 33: Add "(2019)" after Haegeli. […]*

**Response to the review and changes made to the manuscript**

Thank you for pointing this out. We changed the reference accordingly.

*[…] Sterchi and Haegeli (2019) first identified groups of ski runs by clustering similarly coded ski runs over the course of several winter seasons. […]*

**Review**

*[…] Page 6 Line 1: Change "are" to "were" […]*

**Response to the review and changes made to the manuscript**

Thank you for pointing this out. We changed the sentence accordingly.

**Review**

*[…] Page 6 Line 6: Delete "(Sterchi and Haegeli, 2019)". The study has already been referenced. […]*

**Response to the review and changes made to the manuscript**

We believe the reference should stay to be fully clear to what the description is referring.

**Review**

*[…] Page 6 Figure 2 caption: Please confirm whether the Sterchi and Haegeli study is under review or has been published 2019, then update the manuscript accordingly. […]*

**Response to the review and changes made to the manuscript**

Thank you for pointing this inconsistency out. The sentence will included "(Sterchi and Haegeli, 2019)" as the correct reference.

**Review**

*[…] Page 16 Line 12: Typo: "the" should be "they". […]*

**Response to the review and changes made to the manuscript**

Thank you for pointing this out. We changed the sentence accordingly.

---

## Author Response (AR1)

**Exploring the relationship between avalanche hazard and run-list terrain choices at a helicopter skiing operation**

**Author's response**

Reto Sterchi, Pascal Haegeli July 15, 2019

**Dear Margreth Keiler**

Thank you for taking the time to read our manuscript in detail and handle it as editor. We have incorporated the comments brought up by the three referees as well as your own suggested technical items into a revised version of the manuscript.

In addition to these items, we edited the entire manuscript in detail to improve its quality including grammar and rewording of individual sentences to improve readability for the reader.

The following pages of this PDF provide an overview of all changes and include

- a collection of our point-by-point responses to the referees and
- a document including track-changes of our manuscript.

Best regards, Reto Sterchi

**Exploring the relationship between avalanche hazard and large-scale terrain choices at a helicopter skiing operation – Insight from run list ratings**

**Response to Anonymous Referee #1**

Reto Sterchi, Pascal Haegeli July 6 26, 2019

We would like to thank the reviewer for taking the time to read our manuscript in detail and provide constructive feedback. The following sections describe our response to the comments raised by the referee and outline the changes we made to the manuscript to address these concerns.

**1 Title**

**Review (Reviewer #3 made the same comment)**

[...] The tile uses the expression "large-scale"; I recommend the use of "regional" here so that it becomes clear that a large scale (1:10,000 or so) is meant, or "detailed assessment" if this should be the focus, but not – as this expression is quite often also used in NHESS – a nation-wide assessment. [...]

**Response to the review and changes made to the manuscript**

We agree that the expression "large-scale" is not sufficiently specific for describing the spatial scale of our analysis. However, we feel that the proposed "region" is ambiguous as well. Since the scale of the guides' process refers to individual runs, we believe that replacing the expression "large-scale terrain choices" with "run list terrain choices" is most appropriate. To address the reviewer's concern, we made the following changes (highlighted in green):

**Title**

"Exploring the relationship between avalanche hazard conditions and run-list terrain choices at a helicopter skiing operation"

**2 Acceptable risk level**

**Review**

[...] In the abstract as well as in the main text body the authors repeatedly address the term "acceptable risk level", from the overall scientific discussion and concept behind risk and vulnerability, I am wondering what exactly is meant by "acceptable" (death rates below a certain percentage? Number of ski runs without avalanche accident?) and if some explanatory sentences could help here to avoid confusion. [...]

**Response to the review and changes made to the manuscript**

Whereas acceptable avalanche risk levels have explicitly been defined in land use planning (e.g., 1 in 30 years, 1 in 100 years, and 1 in 300 years avalanche risk maps), they have not been defined in backcountry and mechanized skiing. We intended to us the term "acceptable" in a more qualitative way to express that operations do their best to avoid avalanche incidents while acknowledging that the activity is inherently risk and not all of the risk can be eliminated. However, considering this review, we

changed the following instances, where we describe terrain choices intended to reduce the risk to an acceptable level as "appropriate terrain choices" (changes highlighted).

**Page 1, line 10ff:**

[...] Using a large data set of over 25 000 operational run list codes from a mechanized skiing operation, we applied a general linear mixed effects model to explore the relationship between *acceptable* skiing terrain that is deemed appropriate (i.e., status open) and avalanche hazard conditions. [...]

**Page 3, line 3ff:**

[...] The objective of our study is to advance our understanding of the professional avalanche risk management process by quantitatively examining the relationship between acceptable skiing terrain appropriate (i.e., open or closed for guiding) and avalanche hazard conditions at the run scale using historic avalanche hazard assessments and run list ratings from a commercial helicopter skiing operation. [...]

**Page 20, line 1ff:**

[...] For example, explicitly including the likelihood of avalanches and destructive size parameters of the existing avalanche problems in the run list model has the potential to extract more detailed information about the relationship between the avalanche hazard situation and characteristics of runs with acceptable appropriate skiing terrain. [...]

**Page 20, line 11ff:**

[...] Using a large, multi-seasonal dataset of operational run list choices from a mechanized skiing operation, we applied a general linear mixed effects model to quantitatively explore the relationship between avalanche hazard conditions and acceptable appropriate skiing terrain numerically for the first time. [...]

**Page 20, line 29ff:**

[...] For the first time, the effect of avalanche hazard has been isolated from the influence of other factors such as the run list code the day before and the effect of recent skiing. Properly isolating these effects is critical for describing the relationship between avalanche hazard and acceptable appropriate terrain in a meaningful fashion. [...]

**3 Operation vs. Operator**

**Review**

[...] The authors address multiple times the "mechanised skiing operation" but are using data from one operator; maybe the wording could be "mechanised skiing operator" to avoid confusion (e.g., page 1, line 11; page 20, line 11). [...]

**Response to the review and changes made to the manuscript**

The term "operator" usually refers to the actual person that operates (and potentially owns) a mechanized skiing operation. We believe that keeping the term "operation" is more appropriate since our study analyses the run list risk management decisions of an entire organization. We did not make changes to the manuscript.

**4 Illustration of risk management process**

**Review**

[...] On page 2, lines 1-22 the author describe the procedure of assessing avalanche hazard and establishing the run list, it would be useful to underpin this by a Figure showing the different steps by e.g., boxes and arrows in between. [...]

**Response to the review and changes made to the manuscript**

Thank you for highlighting this issue. We believe that a figure will help illustrating the entire process as well as the focus of our study and propose the following figure.

Caption: Hierarchical terrain selection process in mechanized skiing in Canada.

**5 References**

**Review** (Reviewer #3 made the same comment) [...] Please check references for updates, and provide a doi for those references that are in press. [...]

**Response to the review and changes made to the manuscript**

Thank you for highlighting this. The paper is in now press and we updated the reference accordingly.

Walcher, M., Haegeli, P., and Fuchs, S.: Risk of Death and Major Injury from Natural Winter Hazards in Helicopter and Snowcat Skiing in Canada, Wild. Environ. Med., https://doi.org/10.1016/j.wem.2019.04.007, 2019.

**Exploring the relationship between avalanche hazard and large-scale terrain choices at a helicopter skiing operation – Insight from run list ratings**

**Response to Anonymous Referee #2**

Reto Sterchi, Pascal Haegeli July 6, 2019

We would like to thank the reviewer for taking the time to read our manuscript in detail and provide constructive feedback. The following sections describe our response to the comments raised by the referee and outline the changes we made to the manuscript to address these concerns.

**1 Methods: Description of avalanche problems with examples**

**Review**

[...] Page 4, lines 20 to 31: this content does not really belong to the description of the data. In my opinion it also could be skipped. [...]

**Response to the review and changes made to the manuscript**

The avalanche problem types are a crucial part of the Conceptual Model of Avalanche Hazard (CMAH) and the data set used for our study. However, we agree that a brief description of the importance of identifying avalanche problems and their connection to terrain choices might be enough information so that readers can understand what we did in our study and the essence of the results but can refer them to Statham et al (2018) for the details. We shortened and changed the text of lines 20-31 as following.

[...] "While some avalanche problems are of relatively short duration and can be managed easily by avoiding specific terrain features within runs (e.g., wind-loaded slopes when a wind slab avalanche problem is present), others can persist for weeks, even months and require a more conservative risk management approach that includes a broader range of terrain (Haegeli et al., 2010; Statham et al., 2018)." [...]

**2 Methods: Encoding the nature of the ski terrain**

**Review**

[...] Page 5, lines 12 to 30: This part rather belongs to the introduction and could be adapted in a way to emphasise the motivation for this study.

**Response to the review and changes made to the manuscript**

A similar comment was made by reviewer #3. We shortened and changed the text of lines 11-30 as following.

**Page 5, line 11ff**

[...] To identify meaningful patterns between avalanche hazard and terrain choices numerically, it is critical to encode the nature of the available ski runs in a concise, but insightful way. To comprehensively capture of complex nature of entire ski runs into our model in a way that reflects how professional guides perceive them, we used the approach introduced by Sterchi and Haegeli (2019), which groups the ski

runs into operation-specific terrain classes based on multi-seasonal patterns in run list ratings (i.e., revealed terrain preferences). In comparison to existing terrain classification systems with small numbers of universal terrain classes (e.g., ATES; Statham et al., 2006; Campbell and Gould, 2013), Sterchi and Haegeli's approach identifies high-resolution, operation-specific ski run hierarchies based on multiseasonal patterns in run list ratings (i.e., revealed terrain preferences). Sterchi and Haegeli first identified groups of ski runs by clustering similarly coded ski runs over the course of several winter seasons. Subsequently, they arranged the identified groups into a hierarchy that ranges from runs that are almost always open to runs that are only open when conditions are favourable. To better understand the nature of the revealed ski run classes, the authors had a senior lead quide at each participating operation provide a comprehensive but structured description of their ski runs with respect to access, type of terrain, skiing experience, operational role, hazard potential, and guide-ability. Since this ski run classification is based on past operational risk management decisions, it reflects the local terrain expertise and avalanche risk management practices in the context of the available terrain and local snow and avalanche climate conditions (Sterchi and Haegeli, 2019). Thus, this approach represents a more meaningful characterization of ski run classes to analyze professional terrain choices in mechanized skiing operations. [...]

**3 Methods: Avalanche sizes**

**Review**

[...] Page 5, line 18: Better talk about avalanche sizes on figures 1-3 e.g. because the wording has changed in the European classification.

**Response to the review**

Thanks for highlighting this inconsistency in avalanche size description.

**Changes made to the manuscript**

To address the reviewer's concern, we made the following changes (highlighted in green):

[...] and the potential of being seriously injured or deeply buried by avalanches of smaller or equal to size 3. [...]

**4 Methods: Model description**

**Review**

[...] Page 8/9: The explanatory variables and interactions are well explained but could be summarized in a table for a better overview. Further the illustration and explanation of the model is not clear. Better describe model with a formula than with figure 3. Or change Fig.3 for better understanding.

**Response to the review and changes made to the manuscript**

Thank you for pointing that out. After considerable reflection, we believe that a formula would not provide much clarification of the model due to the many variables and interactions involved. However, we believe that structuring the figure in a more table-like layout with additional variable information on could help to overcome the highlighted shortcomings. To address the reviewer's concern, we made the following changes to the figure.

|               | Haz                                                                                                                                            | ard situation                                               |                                                                  |          |
|---------------|------------------------------------------------------------------------------------------------------------------------------------------------|-------------------------------------------------------------|------------------------------------------------------------------|----------|
|               | Variable                                                                                                                                       | Values                                                      | Effect type                                                      |          |
| · > |  <li>Relevant hazard rating</li> <li>Types of avalanche problems present</li> <li>Deep persistent slab</li> <li>Persistent slab</li>  | [0=Low,, 1=Extreme]
[1=present, 0=not present]
[1, 0] | fixed, random
fixed, random
fixed, random                  |          |
| interacted    |  <li>Storm slab</li> <li>Wind slab</li> <li>Cornice</li> <li>Loose wet avalanche</li>                                                 | [1, 0]
[1, 0]
[1, 0]
[1, 0]                        | fixed, random
fixed, random
fixed, random
fixed, random |          |
|               |  <li>Loose Dry avalanche</li> <li>Wet slab</li> <li>Season</li>                                                                       | [1, 0]
[1, 0]
[2013, , 2018]                          | fixed, random
fixed, random
random                         |          |
| Те            | rrain characteristics                                                                                                                          |                                                             | Past use                                                         |          |
| riable        | Values Effect type                                                                                                                             | Variable                                                    | Values                                                           | Effect t |
| Ski run class | s [1, 2, 4, 5, 6] fixed                                                                                                                        | Skied in previous                                           | seven days [1=Yes, 0=No]                                         | fixed    |
|               |                                                                                                                                                |                                                             |                                                                  |          |

**5 Methods: Description of result presentation**

**Review**

[...] Page 10, lines 17 to 24: This section rather fits to the results chapter and explains Fig. 4. [...]

Values

[1=open, 0=closed]

Run list code

previous day

<-----interacted ·-----

**Response to the review and changes made to the manuscript**

Variable

Run list code

We agree that this description of the graph can also be moved into the results section and moved it into section 3.1 where we present figure 4.

**6 Results: Description of parameter estimate**

**Review**

[...] Page 11, line 8: Mention value in the text (e.g. in brackets) for better understanding. [...]

**Response to the review and changes made to the manuscript**

Thanks for pointing out this inconsistency. To address the reviewer's concern, we added the parameter estimates on several instances throughout the results section.

**7 Results: Falsely referenced table**

**Review**

[...] Page 15, line 28: Table 2 not 1 [...]

**Response to the review**

Thanks for highlighting this typo. We made the following changes (highlighted in green):

**Page 15, line 28**

[...] This means that runs in severe alpine terrain are much less likely to be open during times when Deep persistent slab avalanche problems are a concern (OR=0.10 and OR=0.07, respectively, Table 2) [...]

**8 Figures: Size of figure 2**

**Review**

[...] Fig. 2: Is rather small. Could be expanded to entire page width. [...]

**Response to the review and changes made to the manuscript**

A similar comment was made by reviewer #3. We agree with the reviewers and propose to increase the size of the figure and will use the entire width of the page for the figure.